# WEAK-TO-STRONG TRUSTWORTHINESS: ELICITING TRUSTWORTHINESS WITH WEAK SUPERVISION

## ABSTRACT

The rapid proliferation of generative AI, especially large language models (LLMs), has led to their integration into a variety of applications. A key phenomenon known as weak-to-strong generalization – where a strong model trained on a weak model's outputs surpasses the weak model in task performance – has gained significant attention. Yet, whether critical trustworthiness properties such as robustness, fairness, and privacy can generalize similarly remains an open question. In this work, we study this question by examining if a stronger model can inherit trustworthiness properties when fine-tuned on a weaker model's outputs, a process we term weak-to-strong trustworthiness generalization. Specifically, we examine whether a strong model can inherit or even enhance trustworthiness attributes when fine-tuned on a weak model's outputs. To address this, we introduce two foundational training strategies: 1) Weak Trustworthiness Finetuning (Weak TFT), which leverages trustworthiness regularization during the fine-tuning of the weak model, and 2) Weak and Weak-to-Strong Trustworthiness Finetuning (Weak+WTS TFT), which extends regularization to both weak and strong models. Our experimental evaluation on real-world datasets (Adult, OOD Style Transfer, AdvGLUE++, and Enron Emails) reveals that while some trustworthiness properties, such as fairness, adversarial, and OOD robustness, show significant improvement in transfer when both models were regularized, others like privacy do not exhibit signs of weak-to-strong trustworthiness. As the first study to explore trustworthiness generalization via weak-to-strong generalization, our work provides valuable insights into the potential and limitations of this method. Our findings highlight the importance of systematically studying trustworthiness transfer to develop AI systems that are not only accurate but also ethically aligned and reliable in critical applications.

## 1 INTRODUCTION

Over the past few years, there has been a rapid proliferation of generative artificial intelligence (AI), particularly large language models (LLMs) like GPT-3, GPT-4, and their successors. These models have demonstrated remarkable capabilities across a wide range of tasks, including language comprehension (Radford et al., 2019), reasoning (Bubeck et al., 2023) and tabular data generation (Borisov et al., 2023). Their emergent behaviors – unexpected capabilities that arise as models scale – have captured the attention of both academia and industry, leading to widespread adoption and integration into various applications (Wei et al., 2022; Schaeffer et al., 2024).

One intriguing key phenomenon observed in LLMs is known as weak-to-strong generalization (Burns et al., 2024). In this context, a "weak" model, typically smaller or less capable, is used to supervise the training of a larger-sized "weak-to-strong" model. Remarkably, this larger model often surpasses the weak model in performance, even when trained solely on the weak model's outputs. For example, prior research has shown that when a large model is fine-tuned on the predictions of a smaller teacher model for tasks like sentiment analysis or machine translation, the larger model not only learns the task but also generalizes better to new, unseen data (Burns et al., 2024).

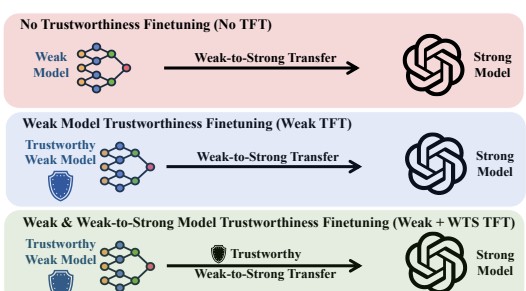

Figure 1: **Weak-to-strong transfer strategies**. We explore the transfer of trustworthiness properties to strong models using different weak-to-strong transfer approaches. **Top**: In *No TFT*, both the weak and WTS models are finetuned solely for task performance. **Middle**: In *Weak TFT*, the weak model is fine-tuned for both task performance and trustworthiness, while the WTS model is finetuned only for task performance. **Bottom**: In *Weak+WTS TFT*, both models are fine-tuned for task performance and trustworthiness.

While performance improvements are valuable, trustworthiness has emerged as a critical aspect of AI systems, especially as LLMs are increasingly deployed in high-stakes domains like healthcare, finance, and legal services (Wang et al., 2023). Trustworthiness encompasses properties such as fairness (avoiding biases against certain groups), privacy (protecting sensitive information), and robustness (maintaining performance under adversarial conditions or distribution shifts). Ensuring these properties is essential to prevent harmful outcomes, comply with regulations, and maintain public confidence in AI technologies. Given the importance of trustworthiness, a vital yet unexplored question arises.

> **Research Question**: Can a strong model inherit or potentially enhance fairness, privacy, and robustness from a weak model through fine-tuning on trustworthy weak model outputs?

In this work, we investigate this critical question by exploring the generalization of trustworthiness properties from weak to strong models, a process we term *weak-to-strong trustworthiness generalization*. We specifically examine whether a strong model can inherit and enhance the trustworthiness properties of a weak model, in addition to improving task performance. To this end, we introduce two novel fine-tuning strategies aimed at enhancing trustworthiness transfer. First, we apply trustworthiness regularization during the fine-tuning of the weak model only. This involves modifying the loss function of the weak model to include fairness, robustness or privacy constraints. We refer to this training strategy as *Weak Trustworthiness Fine-tuning* (Weak TFT). Second, in addition to using a trustworthiness regularized weak model, we also add trustworthiness regularization to the weak-to-strong transfer where we finetune the weak-to-strong model using a trustworthiness regularizer on the trustworthy weak labels. We call this second strategy *Weak and Weak-to-Strong Trustworthiness Fine-tuning* (Weak+WTS TFT). These training strategies are summarized in Figure 1.

We conduct rigorous empirical experiments using the Pythia model suite (Biderman et al., 2023) to evaluate these training strategies on multiple real-world datasets including Adult (fairness), OOD Style Transfer (OOD robustness), AdvGLUE++ (adversarial robustness), and Enron Emails (privacy). Our results demonstrate that while naive fine-tuning of the strong model on the standard weak model outputs only leads to limited weak-to-strong trustworthiness, our proposed training strategies significantly enhance weak-to-strong trustworthiness. Specifically, strong models not only retain but also consistently amplify fairness and robustness properties when both models are regularized. In summary, our contributions can be summarized as follows:

- **Novel Trustworthy Learning Paradigm:** This is the first work to investigate if trustworthiness properties can transfer from a weak to a strong model using weak-to-strong supervision, a process we term *weak-to-strong trustworthiness generalization*.

- **Foundational Training Strategies for Weak-to-Strong Trustworthiness Generalization**: We introduce two baseline training strategies, Weak TFT and Weak+WTS TFT, designed to facilitate weak-to-strong trustworthiness generalization.

- **Weak-to-Strong Trustworthiness Generalization is Feasible**: Our experiments show that some trustworthiness properties can indeed be generalized and even enhanced from weak to strong models.

Our findings provide new insights into systematically transferring and scaling trustworthiness properties from weaker to stronger models. They suggest a viable pathway for developing trustworthy AI

systems without requiring full access to model internals or extensive human supervision. By demonstrating that trustworthiness can be effectively inherited and even enhanced through weak-to-strong generalization, we contribute to the foundational understanding necessary for aligning powerful AI systems with ethical principles.

## 2 RELATED WORK

This work is the first to leverage regularization techniques to study if trustworthiness properties transfer from a weak to a strong model, and one of the first to study weak to strong generalization in large language models. Below we discuss related works for each of these topics.

**Fairness.** Unfair outcomes can arise in language models when they inadvertently encode biases present in the training data, leading to discriminatory practices against certain groups based on sensitive attributes like race, gender, or age (Bolukbasi et al., 2016). Recent efforts to improve fairness in LLMs include data pre-processing, post-processing, and adversarial training such as augmenting training data to balance gender representations (Zhao et al., 2018) and debiasing word embeddings (Huang et al., 2020). Our study sets itself apart by focusing on fine-tuning LLMs using a modified loss function explicitly designed to enhance fairness. Unlike approaches that treat fairness constraints separately or apply post-processing adjustments, we integrate fairness directly into the model's learning objective during fine-tuning.

**Out-of-distribution robustness.** Out-of-distribution robustness describes a model's ability to perform well on inputs that differ from its training distribution. Arora et al. (2021) identify two types of OOD scenarios: 1) semantic shift, where new classes appear at test time, and 2) background shift, where domain or style changes affect the input's presentation without altering core semantics. Various methods aim to enhance OOD robustness, including data augmentation techniques like adversarial perturbations (Madry, 2017; Lecuyer et al., 2019), EDA (Wei & Zou, 2019), and FreeLB (Zhu et al., 2019), as well as training modifications like label smoothing (Szegedy et al., 2016) and focal loss (Lin, 2017). However, recent research has shown that many of these methods do not reliably improve OOD robustness and may even degrade performance on in-distribution tasks; standard fine-tuning often remains a strong baseline (Yuan et al., 2023). In this work, we employ adversarial perturbation as a representative robustness technique. Unlike prior approaches, we focus on generalizing OOD robustness from weaker models to stronger ones, both with and without the use of robustness-enhancing regularization.

**Adversarial robustness.** Machine learning model outputs can be changed by introducing minimal perturbations to a benign input, causing the model to malfunction (Szegedy et al., 2014; Goodfellow et al., 2015; Madry et al., 2018). Several adversarial attack algorithms have been developed that can degrade a large language model's performance on natural language processing tasks such as sentiment analysis, question answering, text classification, and entailment (Jin et al., 2020; Zang et al., 2020; Wang et al., 2020; Li et al., 2020; Garg & Ramakrishnan, 2020). Our work differs from these existing studies and is the first to examine if adversarial robustness properties can transfer from a small model to a larger model trained on the outputs of the small model.

**Model distillation and privacy.** Prior research has explored the use of knowledge distillation as a mechanism to mitigate privacy attacks. One of the most prominent examples is the PATE framework (Papernot et al., 2016), where knowledge distillation is employed to reduce an ensemble of teacher models into a single model with provable privacy guarantees (Dwork et al., 2006). Other works have built on this idea, such as Zheng et al. (2021) and Tang et al. (2022) who construct privacy-preserving model ensembles and then use distillation to consolidate these models. In these approaches, knowledge distillation is often one component of a larger privacy-preserving model, which helps to build models with privacy guarantees. Some research suggests that distillation alone can serve as an effective privacy defense (Shejwalkar & Houmansadr, 2021). Building on this, Mazzone et al. (2022) investigate the use of repeated distillation to protect against membership inference attacks. However, Jagielski et al. (2024) demonstrate through privacy attacks that distilled models without privacy guarantees can still leak sensitive information. In contrast to prior work, our research focuses on the privacy implications of weak-to-strong training, where a large model is trained on the outputs of a smaller model. This approach is the inverse of traditional model distillation, where smaller models are typically trained using the outputs of larger models. Relatively little is known about the privacy

risks and benefits when this process is reversed, making our investigation an important contribution to the field.

## 3 METHODOLOGY

We now present our methodology for investigating the generalization of trustworthiness properties from weak to strong models. Our approach systematically explores whether and how fairness, privacy, and robustness can be effectively generalized from weaker to stronger models. The broader issue we address is: Under what conditions can a weaker model, despite its limitations, most effectively transfer properties such as fairness, privacy, and robustness to a more powerful model? We begin by outlining the weak-to-strong training process, followed by techniques for eliciting specific trustworthiness properties in language models. Finally, we introduce a simple yet effective three-stage training approach that allows us to examine weak-to-strong trustworthiness generalization under different fine-tuning strategies.

### 3.1 PRELIMINARIES

Here we present the key training strategies that underlie our work. First, we discuss how we adapt the weak-to-strong generalization framework introduced by Burns et al. (2024). Following this, we examine widely-used regularization strategies for machine learning models aimed at enhancing trustworthiness properties such as robustness, fairness, and privacy.

**Notation.** We consider training datasets of the form $\{(x_i, y_i)\}_{i=1}^{N}$ where $y_i \in \mathcal{Y}$ is the ground-truth label and $a_i \in \{0, 1\}$ represents a protected attribute (e.g., race or gender) that may be included in the features $x_i$. We denote a classifier $f_\theta : \mathcal{X} \to \mathcal{Y}$ parametrized by $\theta \in \mathbb{R}^d$, mapping inputs $x \in \mathcal{X}$, to labels $\mathcal{Y}$. We define the outputs of a smaller, already trained, fixed classifier $f_w(x)$ as *weak labels*, where $w \in \mathbb{R}^k$ denotes a lower-capacity parameterization than $\theta$ where $k \ll d$. Additionally, let $\ell : \mathbb{R} \times \mathbb{R} \to \mathbb{R}$ represent an appropriate loss function such as cross-entropy loss.

**Weak-to-Strong Training.** In this framework, knowledge transfer from a large pre-trained model occurs by fine-tuning it on the labels produced by a smaller model. This process incorporates an additional auxiliary confidence loss, weighted by $\alpha \in [0, 1]$ that adjusts the confidence in the strong model's predictions relative to the weak labels. This auxiliary loss encourages the strong model to make confident predictions, even when they diverge from the weak labels, potentially enhancing generalization. The loss function is defined as a linear combination of the cross-entropy losses from the weak and strong models: The loss function adapted to our weak-to-strong trustworthiness setting is defined as a linear combination of the losses from the trustworthy weak and strong models

$$\ell_{\text{WTS-AUX}}(x, f_\theta; \alpha, \lambda, f_w) = (1 - \alpha) \cdot \ell\big(f_\theta(x), f_w(x; \lambda)\big) + \alpha \cdot \ell\big(f_\theta(x; \lambda), \hat{f}_{t,\theta}(x)\big), \quad (1)$$

where $f_w(x; \lambda)$ is the fixed trustworthy weak model previously trained with trustworthiness property regularization strength $\lambda$ and $f_\theta(x)$ denotes the strong model. Further, $\hat{f}_{t,\theta}(x)$ represents the hardened strong model predictions according to threshold $t$ that is set proportional to the class weights for each dataset. When $\lambda = 0$, we are in the standard weak-to-strong setting previously studied by Burns et al. (2024). When $\alpha = 0$, we refer to the loss as $\ell_{\text{Naive}}$ since we train on the outputs of the weak model only. In the following, we describe how we obtain the weak trustworthy models $f_w(\cdot; \lambda)$ through various regularization techniques aimed at improving trustworthiness.

**Fairness.** Here we discuss how we can enhance fairness through regularization using a widely-used fairness notion known as Demographic Parity which requires:

$$\mathbb{P}(f_w(x) = 1 | a = 1) = \mathbb{P}(f_w(x) = 1 | a = 0). \quad (2)$$

To enforce this fairness constraint during finetuning, we use the following objective function from Zafar et al. (2017),

$$\min_w \mathcal{L}_{\text{Fair}}(w; \lambda_{\text{Fair}}) = \min_w \frac{1}{N} \sum_{i=1}^{N} \ell(f_w(x_i), y_i) + \lambda_{\text{Fair}} \cdot (a_i - \bar{a}) \cdot f_w(x_i), \quad (3)$$

where $\bar{a} = \frac{1}{N} \sum_{i=1}^{N} a_i$ is the base rate of the protected attribute. The first term in equation 7 encourages to make correct predictions while the second term acts as a fairness regularizer. Specifically,

this term minimizes the covariance between the protected attribute $a_i$ and the model outputs $f_w(x_i)$, encouraging the model to satisfy demographic parity by becoming independent of the protected attribute $a$. The hyperparameter $\lambda_{\text{Fair}}$ controls the tradeoff between prediction accuracy and fairness where a higher value of $\lambda_{\text{Fair}}$ encourages more emphasis on achieving fairer outcomes.

**Adversarial Robustness.** In adversarial training, adversarially perturbed samples are introduced during the training process, enabling the model to learn to become invariant to small input perturbations and thereby become more robust to adversarial attacks. In this setting, the training dataset consists of triplets $(x, x', y)$, where $x$ is a clean input sample, $x'$ is an adversarially manipulated version of $x$ and $y$ is the ground truth label of $x$. The training objective combines the losses from both clean and adversarial samples:

$$\min_w \mathcal{L}_{\text{Adv}}(w; \lambda_{\text{Adv}}) = \min_w \ \frac{1}{N} \sum_{i=1}^{N} (1 - \lambda_{\text{Adv}}) \cdot \ell(f_w(x_i), y_i) + \lambda_{\text{Adv}} \cdot \ell(f_w(x'_i), y_i), \qquad (4)$$

where $\lambda_{\text{Adv}}$ controls the tradeoff between clean and adversarial losses. A higher $\lambda_{\text{Adv}}$ places greater emphasis on robustness to adversarial perturbations.

**Out-of-distribution robustness.** We use embedding perturbations as the method to enhance out-of-distribution robustness, following approaches from Madry (2017); Lecuyer et al. (2019); Zhu et al. (2019). Specifically, we experiment with a setting that adds i.i.d. Gaussian noise to the word embeddings (Bowman et al., 2015; Li et al., 2019). Define $e(x) \in \mathbb{R}^d$ as the word embedding of input $x$, where $d$ is the embedding dimension. We add Gaussian noise $z \sim \mathcal{N}(0, \lambda_{\text{OOD}} \cdot \mathrm{I}_d)$ drawn from a distribution with mean 0 and covariance matrix $\lambda_{\text{OOD}} \cdot \mathrm{I}_d$ to the word embedding which yields a noisy embedding: $\tilde{e}(x; \lambda_{\text{OOD}}) = e(x) + z$. This noisy embedding is then used to finetune the model. Here, let $f_w(x; \lambda_{\text{OOD}}) = g_w(\tilde{e}(x; \lambda_{\text{OOD}}))$ be the output of the language model parametrized by $w$. The objective during finetuning is to minimize the following loss:

$$\min_w \mathcal{L}_{\text{OOD}}(w; \lambda_{\text{OOD}}) = \min_w \frac{1}{N} \sum_{i=1}^{N} \ell\big(y_i, f_w(x_i; \lambda_{\text{OOD}}))\big), \qquad (5)$$

where $\lambda_{\text{OOD}}$ controls the strength of the OOD regularizer. As $\lambda_{\text{OOD}} \to 0$ the model is trained without any regularization, reverting to the vanilla model.

**Privacy.** In $(\lambda_P, \delta)$-differential privacy, the goal is to ensure that the output of an algorithm $\mathcal{A}$ is nearly indistinguishable whether or not any single data point is included in the dataset. Specifically, for any two datasets $D_1$ and $D_2$ that differ by only one element, the algorithm $\mathcal{A}$ satisfies $(\lambda_P, \delta)$-differential privacy if:

$$\mathbb{P}(\mathcal{A}(D_1) \in S) \leq \exp(\lambda_P) \cdot \mathbb{P}(\mathcal{A}(D_2) \in S) + \delta, \qquad (6)$$

for any possible output set $S$. Here, $\lambda_P$ controls the privacy loss, with smaller values indicating stronger privacy guarantees, while $\delta$ allows for a small probability of the privacy guarantee being violated. To operationalize $(\lambda_P, \delta)$-differential privacy, we use the most popular privacy algorithm called DP-SGD as in the work by Abadi et al. (2016). DP-SGD is a variant of classical SGD that comes with privacy guarantees. In summary, the algorithm consists of three fundamental steps: *gradient clipping* with clipping constant $C$, i.e., $\gamma = g(x_i, y_i) \cdot \max(1, C/\|g(x_i, y_i)\|)$ where $g(x_i, y_i) = \nabla_w \mathcal{L}(x_i, y_i)$ is the gradient of the loss function $\ell$ with respect to the model parameters, *aggregation* (i.e., $m = \frac{1}{n} \sum_{i=1}^{n} \gamma_i$) and *adding Gaussian noise* (i.e., $\tilde{m} = m + Y$ where $Y \sim \mathcal{N}(0, \tau^2 \mathrm{I})$ with variance parameter $\tau^2$). By carefully tuning the noise level $\tau^2$, we ensure that the model satisfies the privacy guarantees specified by the parameters $\lambda_P$ and $\delta$.

## 3.2 ELICITING WEAK-TO-STRONG TRUSTWORTHINESS IN LARGE LANGUAGE MODELS

We break the analysis into three stages, each building on the last by varying the regularization applied to the weak and weak-to-strong models.

**No trustworthiness finetuning (No TFT).** In this phase, we establish baseline performance by training the weak, strong, and weak-to-strong models without applying any trustworthiness regularization, following the approach outlined in Burns et al. (2024):

- **Weak model:** We use small, pretrained LLMs as weak supervisors, referred to as weak models. These weak models are finetuned on ground truth labels to generate predictions. Using the

finetuned weak models, we create weak labels by having the weak models make predictions on a held-out validation set.

- **Weak-to-strong transfer:** To evaluate weak-to-strong generalization, we fine-tune a strong model using the weak labels generated by the weak model. This model is referred to as the strong student, and its resulting performance is called the weak-to-strong performance.

**Weak trustworthiness finetuning (Weak TFT).** In this phase, we explore whether a trustworthiness regularized weak model can influence the trustworthiness property of a vanilla strong model trained solely on the output of the trustworthy weak model:

- **Trustworthy weak model:** We use small, pre-trained LLMs as weak supervisors, but unlike in Phase 1, these weak models are fine-tuned on ground truth labels using a trustworthiness regularizer. This regularizer enforces specific trustworthiness properties, such as fairness, privacy, or robustness, during fine-tuning. These models are referred to as trustworthy weak models. Using these models, we generate weak labels by making predictions on a held-out validation set.
- **Weak-to-strong transfer:** To assess whether trustworthiness properties can be transferred from a weak to a strong model, we fine-tune a vanilla strong model using the weak labels generated by the trustworthy weak model.

**Weak and weak-to-strong trustworthiness finetuning (Weak+WTS TFT).** In the final phase, we investigate whether adding trustworthiness regularization to both the weak and weak-to-strong models can further enhance trust transfer (and performance).

- **Trustworthy weak model:** The trustworthy weak model is the same as in Phase 2, where the weak model is fine-tuned on ground truth labels using a trustworthiness regularizer to enforce properties like fairness, privacy, or robustness.
- **Trustworthy weak-to-strong transfer:** In this step, we directly assess how well trustworthiness properties can be transferred from the weak model to the strong model. Unlike in Phase 2, where the strong model was fine-tuned without any regularization, here we finetune the strong model using a trustworthiness regularizer on the weak labels generated by the trustworthy weak model.

## 4 EXPERIMENTAL EVALUATION

In Section 4.1, we empirically evaluate the effectiveness of generalizing trustworthiness properties from a weak to a strong model using the three weak-to-strong training strategies introduced in the previous section. Then, in Section 4.2, we perform a thorough sensitivity analysis, varying the trustworthiness regularization strength, model size, and key hyperparameters specific to weak-to-strong transfer training. We begin by describing the real-world datasets used in our experiments, followed by an overview of the LLMs and relevant baselines used for comparison.

**Datasets.** We evaluate the transfer of trustworthiness properties from small models to large models using four datasets, previously explored by Wang et al. (2023), including the Enron Email dataset (Klimt & Yang, 2004), the Adult dataset (Ding et al., 2021), the OOD Style Transfer dataset (Wang et al., 2023), and the AdvGLUE++ dataset (Wang et al., 2023). For all datasets, we show average results over multiple runs and usually report $\pm 1$ standard deviation across runs.

- **Enron Emails**: This dataset contains over 600,000 emails generated by employees of the Enron Corporation. This dataset includes sensitive personal information, such as email addresses, phone numbers, credit card numbers, and Social Security Numbers, which could be memorized and extracted by language models. For finetuning, we randomly subsampled 10,000 data points.
- **Adult**: This dataset is derived from the 1994 U.S. Census database and contains 48,842 instances with 14 attributes. The task is to classify whether an individual's income exceeds $50,000 (USD) per year. We use the reconstructed Adult dataset provided by Ding et al. (2021) and selected the "sex" feature as the protected attribute to evaluate fairness-related properties.
- **OOD Style Transfer**: This dataset is based on the SST-2 sentiment classification dataset but incorporates a variety of text and style transformations. The transformations (e.g., shifts in language style, vocabulary, syntax, and tone) are applied at both the word and sentence level while preserving the original meaning. For instance, some transformations involve substituting words with Shakespearean equivalents. The task is to correctly classify the sentiment of inputs.

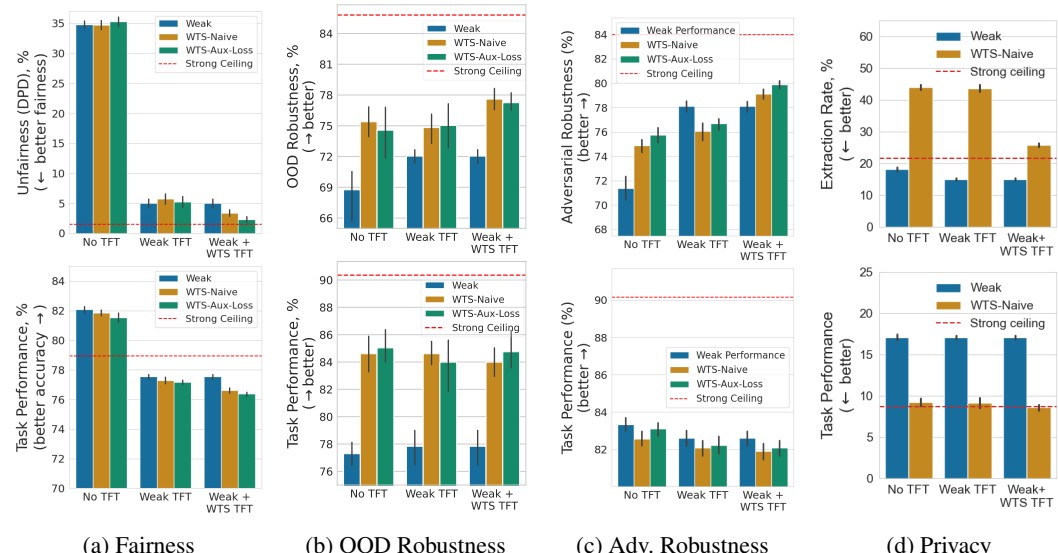

| (a) Fairness | (b) OOD Robustness | (c) Adv. Robustness | (d) Privacy |

Figure 2: **Weak-to-strong trustworthiness for Pythia 14M/410M models**. Trustworthiness properties and task performance for our four properties: Fairness, OOD Robustness, Adversarial Robustness, and Privacy. Note that lower values are better for the top plot in Figure 2a as the y-axis is Unfairness (DPD). Similarly, lower values are better for the top plot in Figure 2d as the the y-axis is Extraction Rate. Results for WTS-Aux-Loss for privacy are omitted since it was the only task involving free data generation, making the auxiliary loss function inapplicable.

- **AdvGLUE++**: This dataset contains clean and adversarial input samples for six NLP tasks: Sentiment analysis (SST-2), duplicate question detection (QQP), multi-genre natural language inference (MNLI, MNLI-mm), recognizing textual entailment (RTE), and question answering (QNLI). It contains around 2K to 15K samples for each of the six tasks. We randomly sample up to 10K samples for each task and aggregate the performance of the model by averaging over these six tasks.

**Large Language Models.** We conduct our experiments using the Pythia model suite (Biderman et al., 2023), which includes models of varying scales (14M, 70M, 410M, 1B). This allows us to systematically explore how model size impacts the effectiveness of trustworthy weak-to-strong generalization. For each model, we finetune on classification tasks by adding a classification head on top of the second-to-last layer. The models are trained using the standard cross-entropy loss.

**Metrics** For *fairness*, we evaluate the finetuned LLMs using the Demographic Parity Difference (DPD) defined as DPD $= \mathbb{P}(f_\theta(x) = 1|a = 1) = \mathbb{P}(f_\theta(x) = 1|a = 0)$. A smaller DPD indicates better fairness, as it reflects minimal disparity in predictions between the two protected groups. For *robustness*, we measure both OOD accuracy and adversarial accuracy, abbreviated as Robust Accuracy (RA), by evaluating the model's performance on OOD and adversarially perturbed test data. Specifically, we compute the RA $= \frac{1}{n_{test}} \sum_{i=1}^{n_{test}} \mathbb{I}[f_\theta(x'_i) = y_i]$, where $x'$ represents either an OOD sample or an adversarially perturbed input, and $\mathbb{I}$ is the indicator function that equals 1 if the prediction is correct. For *privacy*, we evaluate the models using targeted data extraction attacks (Carlini et al., 2021). In this attack, given a prefix sequence and a generated response of $k$ tokens, we compute the extraction rate by determining how many of the $k$-token continuation (suffix) matches the ground truth continuation of the sample. A higher extraction rate indicates a greater risk of private information being memorized and extracted by the model.

**Baselines.** For comparison, we establish reference points for both trustworthiness and task performance. To provide a benchmark, we fine-tune a strong model using ground truth labels with varying levels of trustworthiness regularization. We then select the model that achieves the best trade-off between task performance and trustworthiness. We provide an illustrative example of this selection procedure in Figure A1. This model, referred to as the *strong ceiling*, represents the empirical upper bound of the strong model's capabilities for both task performance and trustworthiness.

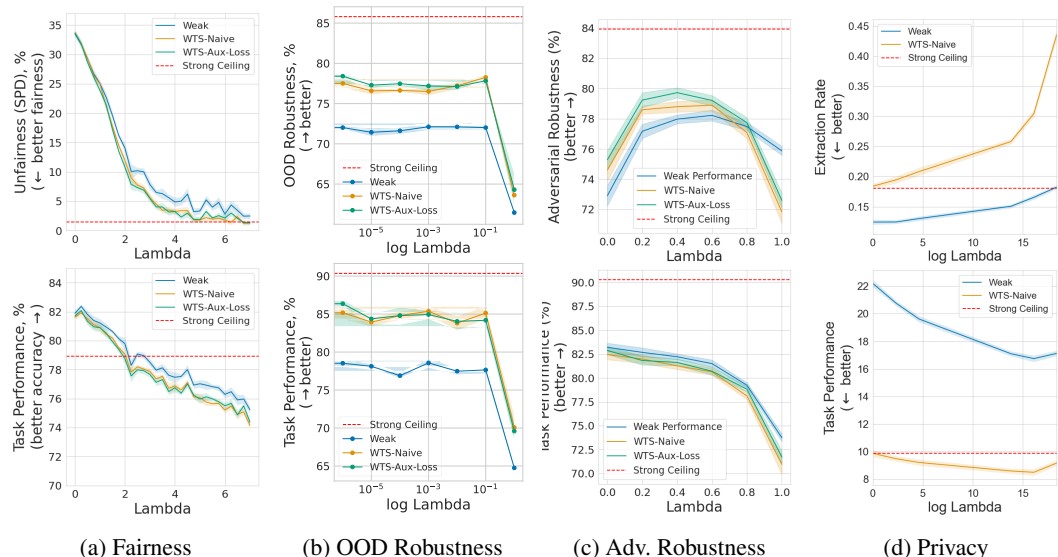

(a) Fairness       (b) OOD Robustness       (c) Adv. Robustness       (d) Privacy

Figure 3: **Varying Lambda for Weak+WTS TFT**. Results for WTS-Aux-Loss for privacy are omitted since it was the only task involving free data generation, making the auxiliary loss function inapplicable.

## 4.1 EVALUATING TRUSTWORTHINESS OF THE WEAK TO STRONG MODEL

We present our results for all four trustworthiness properties across the three phases in Figure 2. For each property, we used Pythia 14M as the weak model and Pythia 410M as the strong model.

**No TFT.** In the initial No TFT phase, where models are trained without any trustworthiness regularization, we expected no clear trustworthy weak-to-strong transfer trends, as no regularization is in place to explicitly enforce trustworthiness properties. Surprisingly, for both OOD and adversarial robustness evaluation we observe that the models demonstrate a trustworthy weak-to-strong trend. Despite the absence of regularization, the stronger models exhibited improved robustness compared to the weaker models, suggesting that some trustworthiness properties may naturally transfer even without explicit constraints. For fairness, we do not observe a trustworthy weak-to-strong trend. The level of unfairness remains constant, regardless of whether we examine the weak model or the weak-to-strong transfer model.

**Weak TFT.** In the Weak TFT phase, regularization is applied to the weak models, which, as expected, improves their trustworthiness in terms of fairness, OOD robustness, and adversarial robustness compared to the No TFT phase. This improvement aligns with our expectations, as the weak models are now being explicitly regularized to enhance their trustworthiness. *The only trustworthiness weak-to-strong trend observed in the Weak TFT phase pertains to OOD robustness.* In this case, the improvement is substantial, increasing robust accuracy from 72% to 75%, representing a gain of 3 percentage points. For all other cases, we do not observe weak-to-strong trends. For privacy, while we do not observe a trustworthy weak-to-strong trend, the gap in extraction rates between the WTS-Naive model and the weak model widens. This widening occurs because the smaller model benefits from privacy regularization, which limits the leakage of information about the training data. In contrast, the WTS-Naive model is trained without any regularization on a second, disjoint dataset, meaning the privacy guarantees from the first model's training phase do not apply. As a result, without explicit regularization during WTS training, the extraction rate for the explicitly regularized weak model decreases much more rapidly, dropping from 18% to 14%. In contrast, the implicitly regularized WTS-Naive model, trained on the outputs of the weak model, experiences a much smaller decline in extraction rate, from 45% to 44%.

**Weak+WTS TFT.** The Weak+WTS TFT phase introduces an additional layer of trustworthiness to the weak-to-strong transfer process, as the training of the WTS model itself is now regularized on top of the existing regularization applied to the weak model in the Weak TFT phase. *With both*

| | Fairness | OOD Robustness | Adv. Robustness | Privacy |
|---|---|---|---|---|
| **No TFT** | ✗ | ✓ | ✓ | ✗ |
| **Weak TFT** | ✗ | ✓ | ✗ | ✗ |
| **Weak+WTS TFT** | ✓ | ✓ | ✓ | ✗ |

Table 1: **Presence of weak-to-strong trustworthiness across trustworthiness properties for different training strategies described in Section 3.2.**

*weak and weak-to-strong models fine-tuned using a trustworthiness regularizer, we observe consistent weak-to-strong trends for all trustworthiness properties, except for privacy.* For fairness, OOD robustness, and adversarial robustness, we observe a statistically significant improvement of each property from weak models to WTS-Naive models. In addition, for fairness and adversarial robustness, there is an enhanced transfer from weak to WTS-Aux-Loss (where WTS-Aux-Loss is more trustworthy than WTS-Naive). For privacy, the extraction rate of the WTS-Naive model decreases by approximately 20 percentage points, dropping from 45% to 26%. This indicates an improvement in privacy compared to the WTS-Naive models from the No TFT and Weak TFT phases, attributed to the explicit regularization applied during WTS-Naive training. However, despite this regularization, the WTS-Naive model remains less private than the weak model, which has an extraction rate of 14%.

**Remarks on the WTS Privacy Trends.** Privacy presents a unique situation. Note that the strong ceiling does not achieve better privacy than the weak model. One reason for this is that privacy is measured with respect to the underlying training dataset (see Appendix C for a more detailed discussion on how the privacy evaluation differs from the evaluations of all other trustworthiness properties). Larger models, all else being equal, tend to memorize more information, leading to a greater risk of private information leakage (Leemann et al., 2024) and as a result larger models are more susceptible to leak private data than small models. Therefore we observe that privacy, as measured by the extraction rate (or membership inference attack success in Figure A11), degrades when transferring knowledge from the smaller model to the larger model, primarily because privacy violations for the WTS model are measured for the larger model, which is more capable of memorizing its training data, rather than the smaller one.

**Tradeoff Between Trustworthiness and Task Performance.** For fairness and adversarial robustness, improvements in trustworthiness come with a slight decline in task performance. However, the decrease in accuracy does not exceed 1.5% across all phases for the two properties while the improvements in trustworthiness were up to 3 percentage points (equivalent to 60% decrease in unfairness). This demonstrates that significant enhancements in trustworthiness can be achieved with minimal sacrifice to task performance.

## 4.2  SENSITIVITY ANALYSIS

In this section, we conduct a comprehensive sensitivity analysis to explore how various parameter values influence the transfer of trustworthiness properties from weak to strong models. Specifically, we examine the impact of different model sizes and the regularization strength $(\lambda_{\text{Fair}}, \lambda_{\text{Adv}}, \lambda_{\text{OOD}}, \lambda_P)$ in the trustworthiness loss functions. We continue the sensitivity analysis for the auxiliary loss weighting parameter $(\alpha)$ used during weak-to-strong transfer in Appendix A. This analysis aims to validate the robustness of our findings from the previous section and to understand the conditions under which weak-to-strong trustworthiness transfer is most effective.

**Sensitivity to Model Size.** To assess the effect of model capacity on trustworthiness transfer, we experimented with different combinations of weak and strong model sizes. We analyzed experiments for four weak/strong configurations: Pythia 14M/410M, Pythia 14M/1B, Pythia 70M/410M, Pythia 70M/1B. Our analysis reveals that the weak-to-strong trends observed in the previous section generally hold across these model sizes for most trustworthiness properties. Specifically, for fairness and OOD robustness, the strong models continued to inherit and, in some cases, enhance the trustworthiness attributes from the weak models across all configurations (Figure A3, Figure A5).

However, we observed a disruption of the weak-to-strong trend for adversarial robustness when using 70M as the weak model. The weak-to-strong trend in adversarial robustness was disrupted in the Weak+WTS TFT phase in the 70M/410M and 70M/1B configurations; the strong models did

not exhibit the expected improvement in adversarial robustness over the weak models (Figure A4). This contrasts with the results from using a 14M weak model, where the strong models did show enhanced adversarial robustness. This disruption suggests that as the weak model becomes more capable, the transfer of adversarial robustness to even stronger models may not follow the same patterns. One possible explanation is that the strong model may already possess sufficient capacity to capture adversarial robustness independently, or the differences in model capacities may affect the dynamics of knowledge transfer. On the other hand, increasing the weak model size from 14M to 70M generally led to improvements in the weak models trustworthiness during the Weak TFT and Weak+WTS TFT phases for both OOD robustness and adversarial robustness. This is expected, as larger weak models have greater capacity to learn complex patterns and trustworthiness properties, providing better supervision for the strong models.

**Sensitivity to Regularization Strength** ($\lambda$)**.** We also investigated how varying the regularization strength in the trustworthiness loss functions affects the transfer of trustworthiness properties. For each property—fairness, robustness, and privacy—we experimented with a range of $\lambda$ values to observe their impact on both the weak and strong models. The trustworthiness weak-to-strong trends described in the previous section maintained across $\lambda$ values in the Weak+WTS TFT phase. The plots of trustworthiness metrics against varying lambda values showed consistent improvements in the WTS-Naive and WTS-Aux-Loss models' trustworthiness attributes when both weak and WTS models were regularized (Figure 3). This consistency suggests that the effectiveness of the Weak+WTS TFT approach is robust to the choice of lambda, provided it is within a reasonable range. Moving from the Weak TFT to the Weak+WTS TFT phase generally made the weak-to-strong trends more apparent across different lambda values. This behavior confirms our analysis in Section 4.1 that weak-to-strong trends are enhanced with increased regularization. Applying trustworthiness regularization to both the weak and strong models amplified the transfer of trustworthiness properties from Figure A2 to Figure 3.

## 5 CONCLUSION

In this paper, we have investigated the critical question of whether trustworthiness properties such as fairness, robustness, and privacy can be transferred from weak to strong models via weak-to-strong generalization. We termed this transfer process weak-to-strong trustworthiness, and introduced two novel approaches aimed at enhancing this transfer. First, Weak Trustworthiness Finetuning (Weak TFT) applies trustworthiness regularization during the fine-tuning of the weak model. Second, Weak and Weak-to-Strong Trustworthiness Finetuning (Weak+WTS TFT) extends this regularization to both the weak and strong models during fine-tuning. Our comprehensive experimental evaluation across real-world datasets reveals that certain trustworthiness properties, namely fairness, adversarial robustness, and out-of-distribution (OOD) robustness, show significant improvement in transfer when both models are regularized. However, we observed that privacy did not exhibit signs of weak-to-strong trustworthiness, highlighting the nuanced nature of transferring different trustworthiness attributes.

**Future Directions.** Our study offers several open avenues for future exploration:

1. *Scope of trustworthiness properties*: While we focused on fairness, robustness, and privacy, other critical trustworthiness aspects such as explainability, accountability, and alignment with human values were not examined. Future work should consider a broader spectrum of trustworthiness attributes to provide a more holistic understanding.

2. *Privacy transfer*: The lack of observed weak-to-strong transfer for privacy suggests that privacy preservation may require fundamentally different approaches. Future research should explore mechanisms that enable privacy properties to transfer or develop new strategies that ensure privacy in the context of weak-to-strong generalization.

3. *Evaluation Metrics*: The metrics used to assess trustworthiness properties may not capture all facets of these complex attributes. Developing more comprehensive evaluation frameworks would provide deeper insights into the models' behavior.

Our work is the first to systematically explore the transfer of trustworthiness properties via weak-to-strong generalization. By emphasizing the potential of this approach, our study provides valuable insights and lays the groundwork for future research in this area.

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

# A  WEAK TO STRONG TRAINING PROCESS

## A.1  TRAINING OBJECTIVE FOR WEAK+WTS TFT

In this section, we give a detailed description of the loss used for the third training strategy presented in Section 3.2.

**Fairness.** To incorporate the fairness constraint into the fine-tuning process, we apply regularization twice yielding the following objective

$$\theta^* \in \arg\min_{\theta} \mathcal{L}_{\text{Fair}}^{\text{WTS}}(\theta; \lambda_{\text{Fair}}^{\text{W}}, \lambda_{\text{Fair}}^{\text{WTS}}, \alpha, f_w)$$

$$= \arg\min_{\theta} \frac{1}{N} \sum_{i=1}^{N} \ell_{\text{WTS-AUX}}(x_i, f_\theta; \alpha, \lambda_{\text{Fair}}^{\text{W}}, f_w) + \lambda_{\text{Fair}}^{\text{WTS}} \cdot (a_i - \bar{a}) \cdot f_\theta(x_i), \tag{7}$$

where $\alpha \in [0, 1]$ is the auxiliary confidence loss weight and where $\bar{a} = \frac{1}{N} \sum_{i=1}^{N} a_i$ is the base rate of the protected attribute. The first term in equation 7 encourages the weak-to-strong model to make correct predictions while the second term acts as an additional fairness regularizer. The hyperparameter $\lambda_{\text{Fair}}^{\text{W}}$ corresponds to the regulrization strength of the weak model while $\lambda_{\text{Fair}}^{\text{WTS}}$ controls the regularization strength for training in this stage.

**Out-of-distribution robustness.** The objective during fine-tuning is to minimize the following loss

$$\theta^* \in \arg\min_{\theta} \mathcal{L}_{\text{OOD}}(\theta; \lambda_{\text{OOD}}^{\text{W}}, \lambda_{\text{OOD}}^{\text{WTS}}, \alpha, f_w)$$

$$= \arg\min_{\theta} \frac{1}{N} \sum_{i=1}^{N} \ell_{\text{WTS-AUX}}(x_i, f_\theta(x_i; \lambda_{\text{OOD}}^{\text{WTS}}); \alpha, \lambda_{\text{OOD}}^{\text{W}}, f_w), \tag{8}$$

where $\alpha \in [0, 1]$ is the auxiliary confidence loss weight. Further, $\lambda_{\text{OOD}}^{\text{W}}$ controls the regularization strength of the fixed weak classifier, while $\lambda_{\text{OOD}}^{\text{WTS}}$ controls the regularization strength of the transfer process. As $\lambda_{\text{OOD}}^{\text{WTS}} = 0$, we are back to our Weak TFT strategy, and as $\lambda_{\text{OOD}}^{\text{WTS}} = \lambda_{\text{OOD}}^{\text{W}} = 0$ the model is trained without any regularization, reverting to the No TFT strategy.

**Adversarial Robustness.** The training objective combines the losses from both clean and adversarial samples:

$$\theta^* \in \arg\min_{\theta} \mathcal{L}_{\text{Adv}}(\theta; \lambda_{\text{Adv}}^{\text{W}}, \lambda_{\text{Adv}}^{\text{WTS}}, \alpha, f_w)$$

$$= \arg\min_{\theta} \frac{1}{N} \sum_{i=1}^{N} (1 - \lambda_{\text{Adv}}^{\text{WTS}}) \, \ell_{\text{WTS-AUX}}(x_i, f_\theta; \alpha, \lambda_{\text{Adv}}^{\text{W}}, f_w) + \lambda_{\text{Adv}}^{\text{WTS}} \, \ell_{\text{WTS-AUX}}(x_i', f_\theta; \alpha, \lambda_{\text{Adv}}^{\text{W}}, f_w), \tag{9}$$

where $\lambda_{\text{Adv}}^{\text{W}}$ controls the regularization strength of the fixed weak classifier, while $\lambda_{\text{Adv}}^{\text{WTS}}$ controls the regularization strength of the transfer process. As $\lambda_{\text{Adv}}^{\text{WTS}} = 0$, we are back to our Weak TFT strategy, and as $\lambda_{\text{Adv}}^{\text{WTS}} = \lambda_{\text{Adv}}^{\text{W}} = 0$ the model is trained without any regularization, reverting to the No TFT strategy.

## A.2  CHOOSING THE HYPERPARAMETERS BASED ON TRADE-OFF CURVES

**Adversarial Robustness.** In this section, we provide an illustrative example of how we selected the parameters for the strong baselines, using adversarial robustness as a case study. We plotted trade-off curves between the trustworthiness properties and task performance, selecting the parameter that corresponds to the optimal trade-off in the top right corner of the Figure A1. We set $\lambda_{Adv}$ for the weak and strong model by independently fine-tuning them on training subset and evaluating on the test subset. We plot original task performance vs. adversarial performance for different values of $\lambda_{Adv}$ and pick the value that offers the best trade-off between clean and adversarial accuracy. Figures A1a and A1b show that $\lambda_{Adv} = 0.3$ achieves good accuracy on original and adversarial samples for both models. Fixing $\lambda_{Adv}$ for the weak model to 0.3, we repeat the same analysis for the weak-to-strong model trained with the naive loss function. Figure A1c shows that $\lambda_{Adv} = 0.3$

offers a reasonable trade-off for the weak-to-strong model as well. Fixing the $\lambda_{Adv}$ parameter to 0.3 for the weak and weak-to-strong models, we vary the $\alpha$ parameter for the auxiliary loss function and plot in figure A1d. We observe that $\alpha = 0.1$ achieves the highest accuracy on both original and adversarial samples. We perform similar analyses for the warm-up period for $\alpha$ and the number of fine-tuning epochs in Figures A1e and A1f and pick the values 0.2 and 6, respectively, for these training parameters.

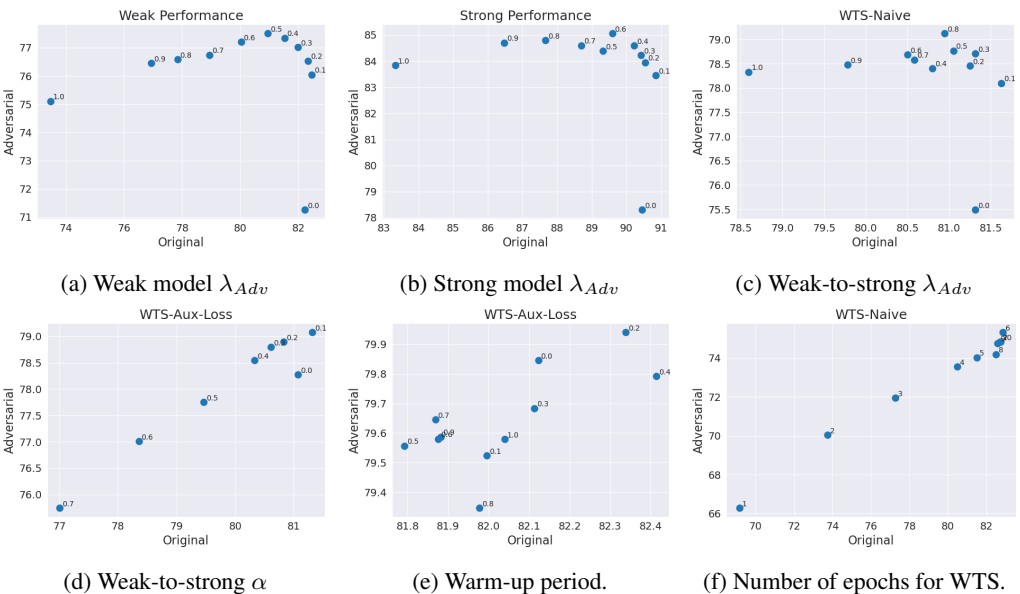

(a) Weak model $\lambda_{Adv}$        (b) Strong model $\lambda_{Adv}$        (c) Weak-to-strong $\lambda_{Adv}$

(d) Weak-to-strong $\alpha$        (e) Warm-up period.        (f) Number of epochs for WTS.

Figure A1: Trade-off between original and adversarial accuracy for different training parameters.

**OOD Robustness.** The standard deviation of the Gaussian Noise is set to $2e - 3$ for both the weak model (Pythia 14M) and the strong model (Pythia 410M). This value was chosen as it allows both models to achieve a balanced tradeoff between OOD robustness and task performance. With the noise standard deviation fixed, we conduct tradeoff experiments by separately adjusting the maximum alpha value for auxiliary loss, the warm-up period, and the number of training epochs. For optimal balance between OOD robustness and task performance, these parameters are set to 0.25, 0.2, and 1, respectively.

## B   DETAILED SENSITIVITY ANALYSIS

**Impact of Size on OOD Robustness.** In this section, we analyze how the sizes of the weak and strong models affect the performance of the weak-to-strong model. We consider two weak model sizes, 14M and 70M, and two strong model sizes, 410M and 1B, resulting in four different experiment configurations. Across all configurations, increasing weak model size consistently leads to noticeable improvements. Increasing the weak model size from 14M to 70M results in significant gains in both OOD robustness and task performance. For example, when comparing the 14M-410M (Figure A5a) and 70M-410M (Figure A5b) configurations, the latter shows enhanced OOD robustness and overall task accuracy. This improvement is even more pronounced when comparing the 14M-1B (Figure A5c) and 70M-1B (Figure A5d) setups. These results suggest that a larger weak model can better capture task-specific patterns, improving both its generalization to out-of-distribution data and its performance on in-distribution tasks, and thus producing more reliable labels for weak-to-strong finetuning.

**Impact of Size on Adversarial Robustness.** In this section, we study the sensitivity of the weak-to-strong trustworthiness fine-tuning to key training parameters like $\lambda_{Adv}$ and $\alpha$. We plot the adversarial robustness and task performance for different values of $\lambda_{Adv}$ and $\alpha$. We observe that adversarial robustness first increases with $\lambda_{Adv}$ and then decreases, achieving a maximum around $0.4$. However,

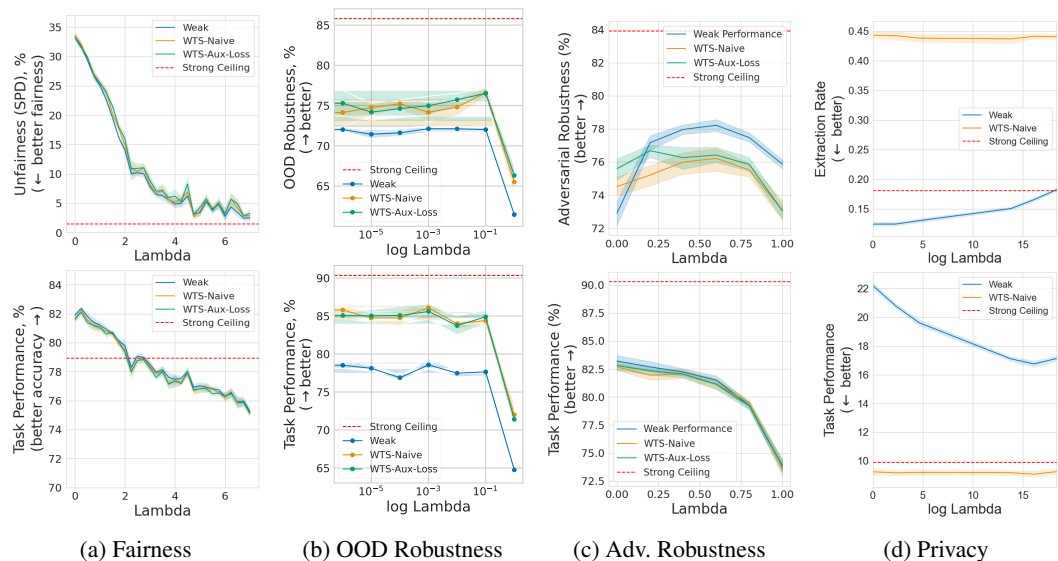

(a) Fairness  (b) OOD Robustness  (c) Adv. Robustness  (d) Privacy

Figure A2: **Varying Lambda for Weak TFT**. Results for WTS-Aux-Loss for privacy are omitted since it was the only task involving free data generation, making the auxiliary loss function inapplicable.

task performance decreases monotonically with $\lambda_{Adv}$. For $\alpha$, the weak-to-strong model performance with auxiliary loss decreases monotonically with the parameter value in all cases.

**Impact of Auxiliary Loss Weighting ($\alpha_{max}$).** The auxiliary loss weighting parameter (maximum alpha) plays a crucial role in balancing the adherence to the weak model's outputs and the strong model's confidence in its predictions. We examined the effect of varying max alpha from 0 to 1 on the performance of the strong models during weak-to-strong transfer. Our experiments showed a degradation of performance with increasing max alpha. As alpha increased from 0 to 1, the performance of the strong models trained with the auxiliary loss (WTS-Aux-Loss) tended to worsen. Higher values of alpha place more emphasis on the strong model's own predictions rather than closely following the weak model's outputs. Therefore, selecting an appropriate value of max alpha is essential to maintain a balance between leveraging the weak model's trustworthiness and allowing the strong model to develop its capabilities. Our results suggest that lower max alpha values are preferable for effective weak-to-strong trustworthiness transfer. For our models, we chose alpha-max values from 0.1 to 0.3.

**Impact of Larger Models (6.9B).** We show that WTS trustworthiness trends are consistent when scaling up the strong model. As referenced in Section 4.2, Figures A3 to A6, show four different weak/strong model size configurations (14M/410M, 70M/410M, 14M/1B, 70M/1B) with consistent property-specific WTS trustworthiness trends holding across model sizes. We also extended our model size sensitivity analysis to include Pythia 6.9B as the strong model for fairness, adversarial robustness, and OOD robustness. The 6.9B model required multiple GPUs to train, and DP-SGD currently does not support multi-GPU computations, so we did not provide 6.9B results for privacy. Figure A9 displays the results and demonstrates similar WTS trustworthiness trends as the previous model configurations. While WTS trustworthiness is inconsistent at the Weak TFT phase, we see consistent WTS trustworthiness at the Weak+WTS TFT phase.

**Impact of Additional Metrics.** We include multiple trustworthiness metrics to further support the WTS trustworthiness trends we observed. In Figure A10, we examine an additional fairness metric: Equalized Odds (True Positive Rate). The consistent WTS trustworthiness trend is maintained across both Demographic Parity and Equalized Odds.

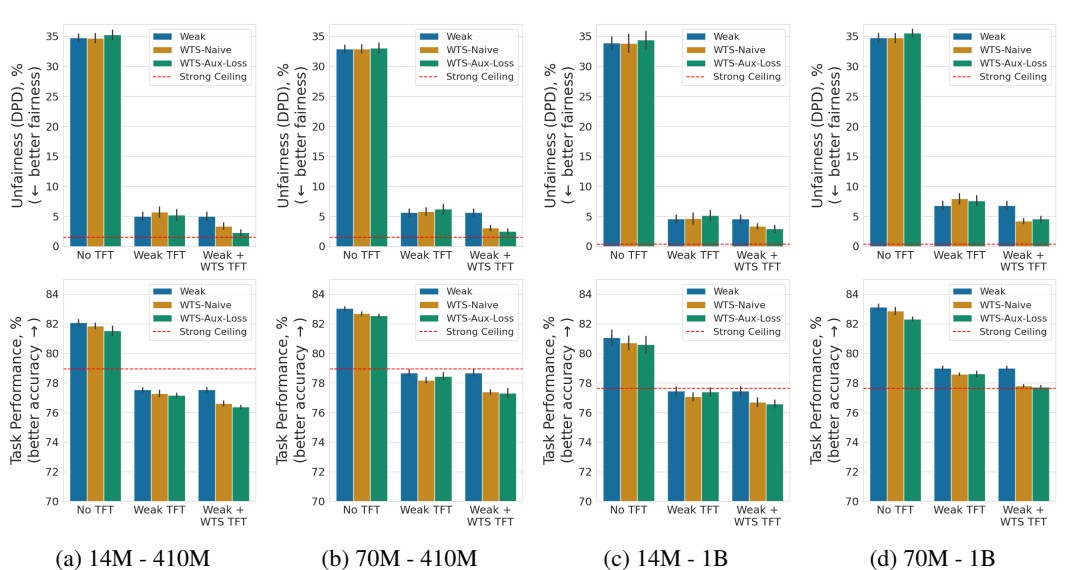

Figure A3: **Varying model size for fairness.**

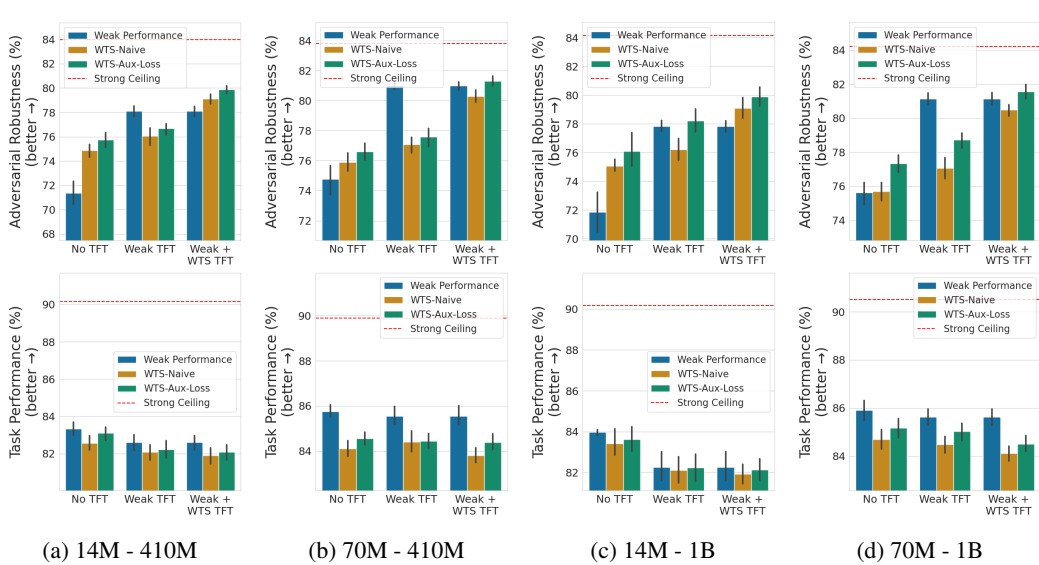

Figure A4: **Varying model size for adversarial robustness.**

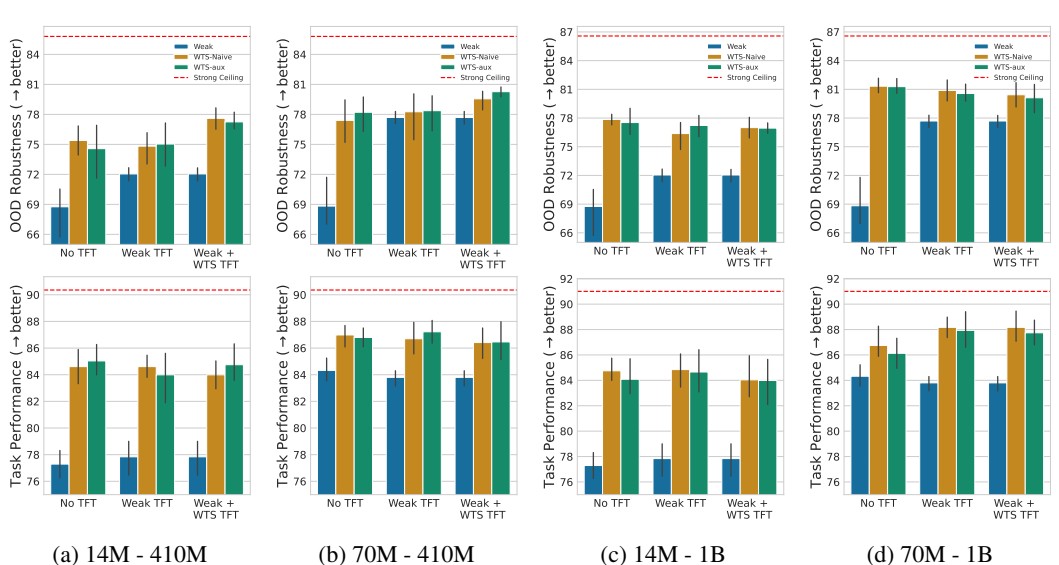

(a) 14M - 410M      (b) 70M - 410M      (c) 14M - 1B      (d) 70M - 1B

Figure A5: **Varying model size for OOD Robustness.**

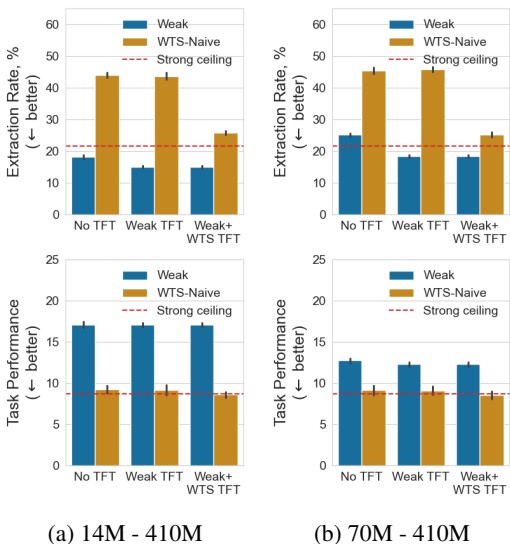

(a) 14M - 410M      (b) 70M - 410M

Figure A6: **Varying model size for privacy.** Due to memory limitations of training models with DP-SGD we did not train the 1B models.

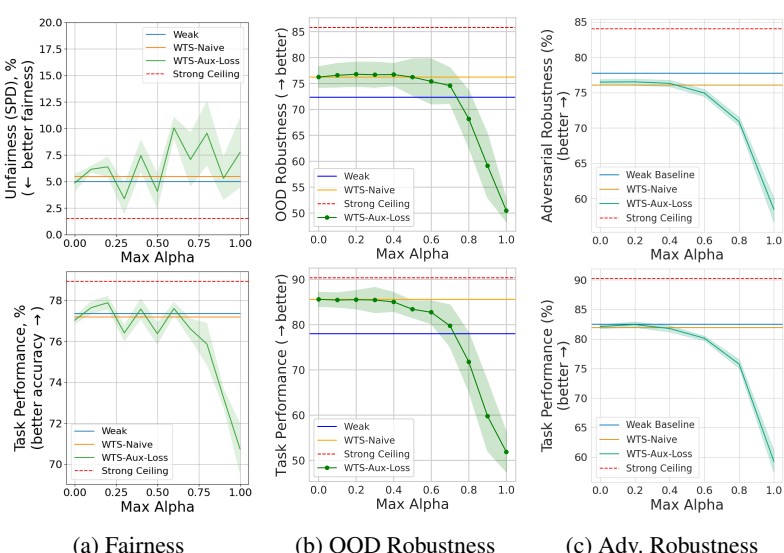

(a) Fairness      (b) OOD Robustness      (c) Adv. Robustness

Figure A7: **Varying Max Alpha for Weak TFT.** Results on privacy are omitted since it was the only task involving free data generation, making the auxiliary loss function inapplicable.

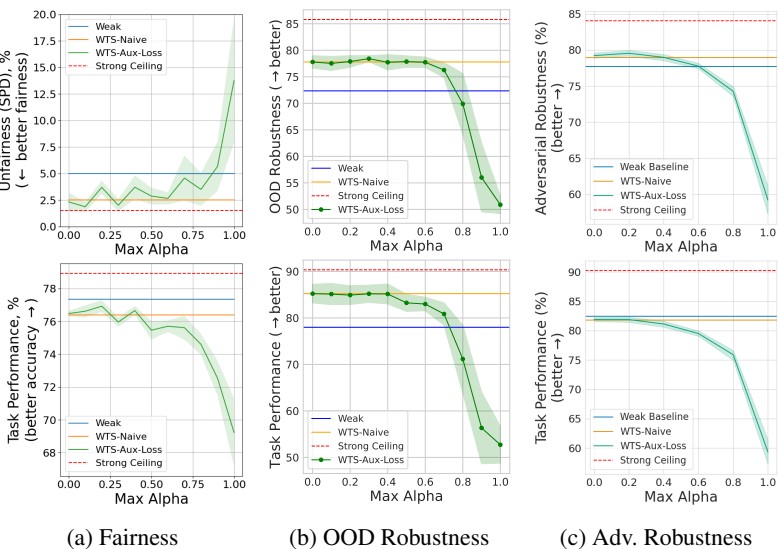

(a) Fairness      (b) OOD Robustness      (c) Adv. Robustness

Figure A8: **Varying Max Alpha for Weak+WTS TFT**. Results for WTS-Aux-Loss for privacy are omitted since it was the only task involving free data generation, making the auxiliary loss function inapplicable.

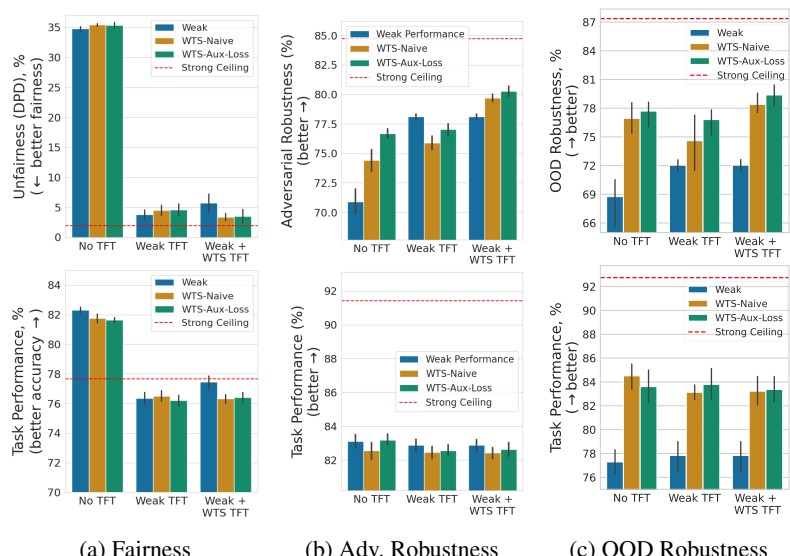

Figure A9: **Model Size Analysis on Pythia 6.9B**. Results for model size sensitivity with Pythia 14M as the weak model and Pythia 6.9B as the strong model for fairness, adversarial robustness, and OOD robustness properties. We see that the WTS trends we identified earlier are maintained for the larger strong model.

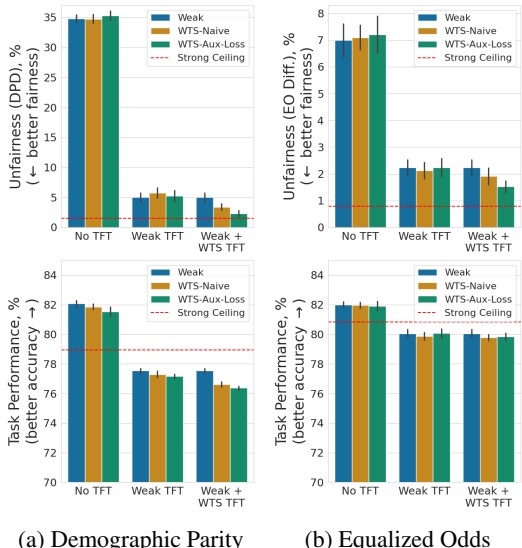

Figure A10: **Sensitivity to Fairness Metrics**. Side-by-side results for two fairness metrics: Demographic Parity and Equalized Odds (True Positive Rate). The WTS trustworthiness trend is maintained across both metrics.

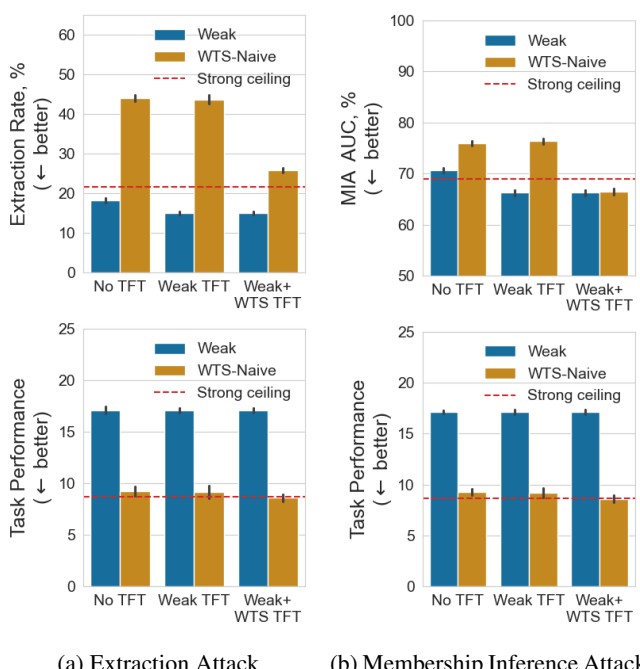

(a) Extraction Attack          (b) Membership Inference Attack

Figure A11: **Sensitivity to Privacy Metrics**. Side-by-side results for two privacy metrics: Extraction Attack and Membership Inference Attack. In both cases, we do not observe weak-to-strong trustworthiness trends.

## C  DATASET AND EVALUATION DETAILS

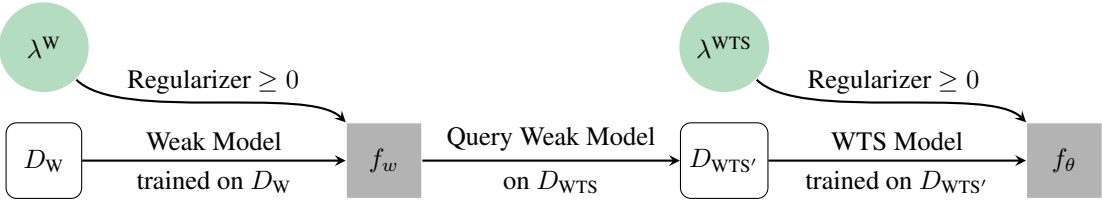

(a) **Model training overview**. The weak model $f_w$ is trained on $D_\text{W} = \{(x_i, y_i)\}$. Subsequently, we use the weak model $f_w$ to label the weak-to-strong training dataset $D_\text{WTS} = \{(x_i, y_i)\}$ resulting in $D_{\text{WTS}'} = \{(x_i, f_w(x_i))\}$. We use $D_{\text{WTS}'}$ to train the weak-to-strong model $f_\theta$.

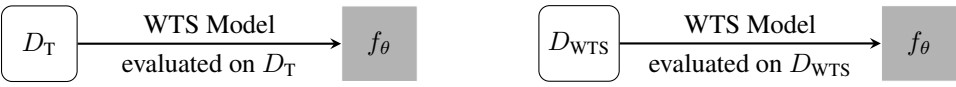

(b) **Trustworthiness property evaluation.** Typically, the trustworthiness properties for the WTS model are evaluated on a separate test set $D_\text{T}$.

(c) **Privacy Leakage Evaluation.** The privacy leakage for the WTS model is evaluated using the ground truth train set $D_\text{WTS}$.

Figure A12: **Data usage during training and evaluation.** In Figure A12a, we describe which data is used to train the weak and the weak-to-strong models, while Figures A12b and A12c describe which data is used for evaluation.

### C.1  DATA USAGE DURING TRAINING AND EVALUATION

Figure A12 describes which data is used for both training the weak and the WTS models as well as for evaluation of the WTS model.

**Data used to train the WTS model.** The weak model $f_w$ is trained on the labeled dataset $D_\text{W} = \{(x_i, y_i)\}$. Once trained, we use the weak model $f_w$ to label the weak-to-strong training dataset $D_\text{WTS} = \{(x_i, y_i)\}$ resulting in $D_{\text{WTS}'} = \{(x_i, f_w(x_i))\}$. We use $D_{\text{WTS}'}$ to train the weak-to-strong model $f_\theta$. Notably, there is no overlap between $D_\text{WTS}$ and $D_\text{W}$.

**Trustworthiness Evaluation.** We evaluate the trustworthiness properties adversarial robustness, OOD robustness as well as Demographic Parity and Equlaized Odds for all models (weak model, WTS model and strong ceiling) on the same held out test set for the respective problem. For privacy, we evaluated the trustworthiness properties of the weak and the strong model on their training set $D_\text{W}$ while the privacy leakage for the WTS model is evaluated on $D_\text{WTS}$. For privacy considerations, we evaluated the trustworthiness properties of the weak and strong models on their training set D $D_\text{W}$, while the privacy leakage for the WTS model is assessed on $D_\text{WTS}$.

### C.2  ADDITIONAL ADVERSARIAL ROBUSTNESS DATASET DETAILS

In this section, we evaluate the adversarial robustness of the weak-to-strong models and compare with the weak baseline and the strong ceiling. We use Pythia 14M as the weak model and Pythia 410M as the strong model. We create training, holdout and test subsets of the AdvGLUE++ dataset using 40%, 40% and 20% of samples, respectively, from each task in the dataset. We use the training subset to fine-tune our models to be adversarially robust. We use the holdout subset to generate labels from the weak model to be used in the weak-to-strong training process. To evaluate the clean and adversarial accuracy of our models, we evaluate them on a test subset of the AdvGLUE++ dataset and average the performance across the six NLP tasks in this dataset.

In particular, to evaluate weak-to-strong trends in adversarial robustness, we use the AdvGLUE++ dataset (Wang et al., 2023), an extension of the AdvGLUE dataset (Wang et al., 2021). AdvGLUE++ is a comprehensive benchmark designed to test adversarial robustness across multiple natural language processing (NLP) tasks and adversarial attack algorithms. This dataset includes

adversarial examples for six widely used NLP tasks, each representing a distinct domain or linguistic challenge. The Stanford Sentiment Treebank (SST-2) task involves sentiment analysis, requiring the classification of sentences as having a positive or negative sentiment. The Quora Question Pairs (QQP) task identifies whether two questions convey the same meaning. The Multi-Genre Natural Language Inference (MNLI) task requires reasoning about entailment, contradiction, or neutrality between pairs of sentences. It includes a mismatched variant, MNLI-mm, where validation and test data originate from out-of-domain sources, increasing the challenge of generalization. The Question-answering NLI (QNLI) task is framed as an entailment problem between a question and an answer candidate. The Recognizing Textual Entailment (RTE) is a binary entailment task that aims to determine whether the meaning of one text can be inferred from another.

Adversarial examples in AdvGLUE++ are generated using a variety of attack algorithms, each representing a distinct perturbation strategy. TextBugger introduces typo-based perturbations that minimally alter characters while preserving the utility of benign text. TextFooler generates embedding similarity-based perturbations by substituting words with contextually plausible alternatives. BERT-ATTACK leverages BERT's language modeling capabilities to create context-aware adversarial samples. SememePSO relies on semantic representations and combinatorial optimization to generate knowledge-guided perturbations. SemAttack employs semantic optimization-based techniques by manipulating various semantic spaces to produce natural-looking adversarial texts.

The experimental results for adversarial robustness are presented as aggregated accuracy values across all six tasks and five attack algorithms. This approach enables us to evaluate the weak-to-strong trends in a comprehensive and robust manner. The results show that our findings are consistent across a wide range of NLP tasks and adversarial attacks, indicating that they are not influenced by the specific characteristics of any single setting.

### C.3 ADDITIONAL OOD DATASET DETAILS

We use the same OOD data created by Wang et al. (2023). For ID data, we use the original SST-2 dataset but exclude the samples that are source samples for creating the OOD data. We split the ID data into training, validation, and heldout subsets. Specifically, 50% of the ID data is allocated for training and validation, where 95% of that portion is used for training and the remaining 5% is for validation. The other half represents the held-out data that is used for generating labels from the weak model for weak-to-strong finetuning. For evaluation, we use the in-distribution validation samples to measure ID performance and the OOD test samples to obtain OOD performance.

