# OpenReview forum: "Weak-to-Strong Trustworthiness: Eliciting Trustworthiness with Weak Supervision"
_ICLR.cc/2025/Conference — Submitted to ICLR 2025_

### Official Review · Reviewer_CUwC · 2024-10-31

**Soundness:** 2
**Presentation:** 2
**Contribution:** 1
**Rating:** 3
**Confidence:** 4

**Summary:**

This paper conducts an empirical study of weak-to-strong generalization for trustworthiness (fairness, OOD generalization, adversarial robustness and privacy). They compared three ways to perform weak-to-strong generalization, where the difference is whether the trustworthiness regularization is added to the loss of the weak/strong models. Experiments show the Weak+WTS TFT approach is the most effective way of transferring trustworthiness.

**Strengths:**

- This work considers 4 different aspects of trustworthiness.
- It performs a relatively comprehensive study of parameter sensitivity.

**Weaknesses:**

- The motivation of the weak-to-strong transfer in this work is questionable. It can be meaningful to study this problem in some special cases, such as when the labeled dataset is small while there exists a large unlabeled dataset. If the strong model is only trained with the small labeled data, it can suffer from underfitting, thus making it more meaningful to train the weak model first and train the strong model with weak labels generated by the weak model. However, this work does not consider such cases.
- This work is benchmarking three different ways to fine-tune the strong model, there is not much technical novelty.

**Questions:**

1. In figure 2 (d), why does the Weak+WTS TFT perform better than the strong ceiling?

---

> ### Author Response · Authors · 2024-11-21
> **Response to Reviewer CUwC (Part 1)**
>
> We thank the reviewer for their clarifying questions and thoughful comments and believe that they help improving the communication of our results. Below we address individual points raised by the reviewer.
>
> > The motivation of the weak-to-strong transfer in this work is questionable.
>
> We agree that the motivation from weak-to-strong trustworthiness might differ from the motivation of the OpenAI work. While there are many motivations for our work, below we have compiled a short list of 6 use-cases in which weak-to-strong trustworthiness is useful.
>
>
>
> 1. **Data Privacy and Confidentiality Constraints**
>     - **Regulatory Compliance**: In sectors like healthcare and finance, strict regulations (e.g., HIPAA, GDPR) protect sensitive data. Trustworthiness properties such as privacy, robustness, and fairness are crucial for models in these sectors (e.g. algorithmic fairness in patient treatment predictions, adversarial robustness of loan approval models). Organizations may be prohibited from sharing raw data or labels, even internally or with trusted partners. A weak model trained by a group with access to sensitive data can generate outputs that abstract away personal identifiers, allowing a stronger model from other institutions to learn without accessing the protected ground truth labels.
>     - **Anonymized Data Sharing**: When data anonymization is insufficient to meet privacy standards, organizations can share model outputs from (privatized) weak models instead of data. The strong models learn from these outputs, inheriting trustworthiness properties without compromising privacy.
>
> 2. **Cross-Organizational Collaboration Without Data Exchange**
>    - **Collaborative Learning**: Companies or institutions might collaborate to improve AI models but cannot share proprietary or sensitive data due to competitive concerns or legal restrictions. By training a strong model on the outputs of a weak model from a partner organization, they can enhance trustworthiness and performance while respecting data ownership boundaries.
>    - **Knowledge Transfer in Mergers and Acquisitions**: During corporate restructuring, legal constraints may delay the transfer of data assets. However, models and their outputs might be shareable, allowing the development of improved, trustworthy models in the interim.
>
> 3. **Limited or Expensive Access to Ground Truth Labels**
>    - **High Annotation Costs**: In domains requiring expert knowledge for labeling (e.g., medical imaging, legal document analysis), obtaining ground truth labels is costly and time-consuming. A weak model trained on a limited labeled dataset can generate pseudo-labels for a larger unlabeled dataset. The strong model can then learn from these pseudo-labels to improve performance and trustworthiness.
>    - **Dynamic Data Environments**: In rapidly changing fields like healthcare and finance, ground truth labels may become outdated quickly. Along with the Regulatory Compliance use case, another institution can continuously learn from labels from an updating weak model (from a group with access to sensitive data) even with changing ground truth labels.

---

> ### Author Response · Authors · 2024-11-21
> **Response to Reviewer CUwC (Part 2)**
>
> > This work is benchmarking three different ways to fine-tune the strong model, there is not much technical novelty.
>
>
> While our paper does not propose a new algorithm, we argue that empirical insights into the capabilities and limitations of large language models (LLMs) can be as impactful as methodological innovations, especially given the rapid advancements in LLM capabilities.
>
> In particular, our work represents the first comprehensive exploration of Trustworthy Weak-to-Strong Generalization (TWTS), examining how trustworthiness properties such as fairness, robustness, and privacy can be effectively transferred from weak to strong models. By introducing this novel evaluation paradigm and applying it across diverse trustworthiness criteria, we provide the community with foundational insights into a critical, underexplored problem space. This work is intended to establish a robust baseline and raise awareness of TWTS as a valuable research direction, paving the way for follow-up studies.
>
> Importantly, our two suggested methods - Weak TFT and Weak+WTS TFT - go beyond standard weak-to-strong transfer by incorporating trustworthiness regularization during different stages of training. These methods establish two strong baselines for trustworthy generalization that future research will need to surpass. This is particularly significant because our work not only evaluates existing techniques but actively provides new benchmarks to drive progress in this area. These baselines help anchor the trustworthiness generalization problem and will serve as key reference points for the community going forward.
>
> Furthermore, the importance of empirical work has been underscored by several prominent papers that had limited methodological innovation but were celebrated for their insightful empirical evaluations. Examples include:
> -  Weak-to-Strong Generalization: Eliciting Strong Capabilities With Weak Supervision, ICML 2024; https://arxiv.org/abs/2312.09390
> -  DecodingTrust: A Comprehensive Assessment of Trustworthiness in GPT Models, NeurIPS 2023; https://arxiv.org/abs/2306.11698
> -  Chain-of-Thought Prompting Elicits Reasoning in Large Language Models, NeurIPS 2022; https://arxiv.org/abs/2201.11903
> - Rethinking the Role of Demonstrations: What Makes In-Context Learning Work?; EMNLP 2022; https://arxiv.org/abs/2202.12837
>
> These works have shown that impactful research need not always hinge on technical novelty; instead, carefully designed empirical studies addressing important gaps in understanding can be equally valuable to the community. Similarly, our work contributes by not only introducing TWTS as a research direction but also providing rigorous, well-defined baselines that set the stage for future advancements in trustworthy generalization.
>
> We hope this response clarifies the significance of our contributions and the rationale for focusing on this empirical investigation.
>
> > In figure 2 (d), why does the Weak+WTS TFT perform better than the strong ceiling?
>
> The one standard error confidence interval for Weak+WTS TFT’s includes the value of the strong ceiling, so it is not significantly better than the strong ceiling.
>
>
> ----
> We thank the reviewer again for their thoughtful comments and feedback. We hope we addressed all your questions/concerns/comments adequately. In light of our clarifications, please consider increasing your score.
>
> ----

---

> > ### Comment · Reviewer_CUwC · 2024-11-24
> >
> > 1. I tried to understand the motivation explained by the authors, but still cannot get it. Could the authors provide a single very solid example when this weak-to-strong generalization setting is useful in practice? Why is the weak model even needed in any of the cases?

---

> > > ### Author Response · Authors · 2024-11-25
> > > **Response to Reviewer CUwC (Part 1)**
> > >
> > > Thank you for your response and for engaging in the discussion with us. We very much appreciate this.
> > >
> > > Below we outline a comprehensive example, highlighting the practical motivation for trustworthy weak-to-strong (TWTS) generalization and the importance of the weak model.
> > >
> > > **Setup**: There are two organizations: Hospital A and Research Lab B. Hospital A is a regional medical center with extensive records of sensitive patient data. Research Lab B is a leading AI research institution with computational resources, advanced models, and employee expertise. Due to strict privacy regulations for protected health information, Hospital A cannot provide Research Lab B access to its complete medical dataset.
> > >
> > > **Challenges**. Hospital A and Research Lab B face many challenges in developing a strong model.
> > > - **Data Privacy Constraints**: Hospital A cannot share raw patient data or labels with external entities such as Research Lab B due to privacy regulations and ethical obligations. Without the information from Hospital A’s data (through weak supervision), Research Lab B cannot train high-performing trustworthy models.
> > > - **Resource Limitations**: Hospital A can train a trustworthy weak model but lacks the computational infrastructure to train a large-scale model (strong model).
> > > - **Need for Trustworthiness**:
> > > The weak model developed by Hospital A cannot ensure high enough accuracy and trustworthiness concurrently. It is especially important to ensure trustworthiness for medical models due to the high-stakes decisions from outcome severity and magnitude of people affected. It is crucial for these models to be fair across diverse demographics, compliant with privacy standards and robust against adversarial attacks
> > >
> > >
> > > **Trustworthy Weak to Strong Generalization Resolves these Challenges**: Applying trustworthy weak-to-strong generalization helps Hospital A and Research Lab B achieve their objective as follows. Our paper provides a comprehensive evaluation of Trustworthy Weak-to-Strong Generalization, demonstrating how trustworthiness properties can be transferred and enhanced through various training strategies, all while allowing access to only the weak outputs and not ground truth labels for the larger model.
> > >
> > > 1.  **Training the Weak Model at Hospital A**:
> > >     - Hospital A trains a weak model on its sensitive data, incorporating trustworthiness constraints (e.g., fairness metrics across patient groups, adversarial robustness, etc.).
> > >
> > > 2. **Transferring Knowledge Without Sharing Sensitive Data:**
> > >     - Hospital A shares the outputs of the weak model (not the raw data or labels) with Research Lab B. This approach complies with privacy regulations, as the outputs do not contain personally identifiable information.
> > >     - Then, Hospital A’s weak model can produce outputs on a set of non-sensitive data or publicly available data points which Research Lab B has access to.
> > >
> > > 3. **Training Strong Model at Research Lab B via Trustworthy Weak-to-Strong Generalization**:
> > >     - Research Lab B uses its advanced infrastructure and computational resources to train a strong model, utilizing Hospital A’s weak model's outputs.
> > >     - By incorporating trustworthiness regularization during fine-tuning (appropriate strategies for different properties outlined in our paper), the strong model aims to inherit and enhance the trustworthiness properties of the weak model.
> > >
> > > 4. **Deploying the Trustworthy Strong Model**:
> > >    - Research Lab B’s strong model is deployed back to Hospital A, improving diagnostic performance while achieving greater trustworthiness.
> > >
> > > **On the Relevance of the Weak Model**: The weak model is needed in this case to enable knowledge transfer under privacy constraints and overcome Hospital A’s resource limitations. The weak model acts as a bridge, allowing Hospital A to contribute valuable insights from its sensitive data without violating privacy laws while collaborating with Research Lab B. The weak model incorporates specific trustworthiness constraints relevant to Hospital A's patient population, which are then transferred to Research Lab B’s strong model. Through trustworthy weak-to-strong generalization, Hospital A can participate in developing advanced AI solutions without needing extensive computational resources or violating privacy regulations.

---

> > > > ### Author Response · Authors · 2024-11-25
> > > > **Response to Reviewer CUwC (Part 2)**
> > > >
> > > > **Practical Impact of Trustworthy WTS Generlization**: We highlight the following practical impacts of trustworthy weak-to-strong generalization:
> > > >
> > > > - **Regulatory Compliance**: TWTS generalization adheres to legal frameworks such as HIPAA and GDPR, allowing for cross-institution collaboration on developing strong models in high-stakes settings with sensitive data.
> > > > - **Improved Patient Outcomes**: Hospitals and research institutions develop a high-performance model that is also trustworthy, enhancing patient care quality and accessibility.
> > > > - **Scalability**: TWTS generalization can be implemented across multiple hospitals, each contributing to and benefiting from the strong model without compromising data privacy. By developing a centralized strong model, hospitals that lack resources to train their own AI models can access advanced AI capabilities without sharing sensitive medical records.
> > > >
> > > > ----
> > > > Please let us know if you have any further questions.

---

> > > > > ### Comment · Reviewer_CUwC · 2024-11-25
> > > > >
> > > > > I got your point. But in your work it seems you did not consider such cases in your manuscript or experiments. For example, training multiple weak models on different data sources and training one strong model by KD from the small models. In addition, there's no much empirical results on the privacy protection side.

---

> > > > > > ### Author Response · Authors · 2024-11-25
> > > > > > **Response to Reviewer CuwC**
> > > > > >
> > > > > > While the multiple model direction is interesting, we respectfully maintain that our work's scope and contributions are significant as presented. Our research deliberately focuses on fundamental challenges in high-stakes ML applications where trust and reliability are paramount. We systematically explore four critical axes that directly impact ML deployment in critical applications:
> > > > > >
> > > > > > - Fairness: Ensuring equitable model performance across demographic groups;
> > > > > > - Out-of-Distribution Robustness: Reliable behavior on shifted data;
> > > > > > - Adversarial Robustness: Resilience against malicious attacks;
> > > > > > - Privacy: Protection of sensitive training data.
> > > > > >
> > > > > > While the suggested experiments with multiple weak models are interesting directions, we view these as important follow up questions that will build upon our core findings rather than the fundamental challenges that needed to be addressed in this work.
> > > > > >
> > > > > > Regarding privacy protection results, we respectfully disagree with the assessment of insufficient empirical validation. Our experiments in Section 4 and Appendix B provide comprehensive privacy evaluations, including extraction and membership inference attacks, following current standards in privacy-preserving ML literature. Importantly, our work makes a significant contribution by demonstrating that TWTS Generalization does not work for privacy (with detailed explanations in Section 4.1). This negative result is valuable to the community and identifies an important open challenge: understanding the specific conditions under which private WTS is possible.
> > > > > >
> > > > > > We believe our focused investigation of these four fundamental axes provides a strong foundation for future work, including the interesting directions the reviewer suggests.

---

> ### Comment · Reviewer_CUwC · 2024-11-25
>
> Note that the authors mentioned that
> > Regulatory Compliance: TWTS generalization adheres to legal frameworks such as HIPAA and GDPR, allowing for **cross-institution collaboration** on developing strong models in high-stakes settings with sensitive data.
> Improved Patient Outcomes: Hospitals and research institutions develop a high-performance model that is also trustworthy, enhancing patient care quality and accessibility.
> Scalability: TWTS generalization can be implemented across **multiple hospitals, each contributing to and benefiting from the strong model without compromising data privacy**. By developing a centralized strong model, hospitals that lack resources to train their own AI models can access advanced AI capabilities without sharing sensitive medical records.
>
> However, in the paper, the generalization is from a single weak model to a strong model. Therefore, there does exist a mismatch between the motivation mentioned by the authors in the rebuttal and the method and experiment presented in the paper.

---

> > ### Author Response · Authors · 2024-11-25
> > **Response to Reviewer CUwC**
> >
> > Thre reviewer asked for ''a single very solid example'' which we provided in Part 1 of our response above. This example involves a Hospital A and Research Lab B, and **does not involve multiple hospitals**.
> >
> > In part 2 of our response above we provided *Practical Implications of Trustworthy WTS Generlization* and where it can be useful.
> > While the experiments in our work cover the first two use cases *Regulatory Comliance* and *Improved Patient Oucomes*, another use cases can involve multiple hospitals. We view this case as an interesting follow up questions that will build upon our core findings rather than the fundamental challenges our work addressed.

---

> > > ### Comment · Reviewer_CUwC · 2024-11-25
> > >
> > > Okay, following the authors' argument, let's consider the case where there is only a single weak model. My question is, why cannot we train the strong model on this data? Does there exist any regulation that requires people to train a weak model on the original data? This is the reason needed for the authors to justify their approach that utilizes knowledge distillation to train the strong model. To the best of my knowledge, I have not heard of any regulation like this.

---

> > > > ### Author Response · Authors · 2024-11-25
> > > > **Response to Reviewer CuwC**
> > > >
> > > > We clarify our example as follows, highlighting why the strong model cannot be directly trained on the data.
> > > >
> > > > **Regulatory Constraints Prevent Hospital A from Sharing Raw Data**: Regulations such as **HIPAA** (Health Insurance Portability and Accountability Act) in the U.S. and **GDPR** (General Data Protection Regulation) in the EU **explicitly prohibit the sharing of protected health information** (PHI) without patient consent. Hospital A cannot legally transfer raw patient data or labels to any external entity, including Research Lab B.
> > > >
> > > > **Hospital A Unable to Train Strong Model Itself**: Due to resource limitations, Hospital A lacks the computational infrastructure to train a large-scale strong model.
> > > >
> > > > **Role of the Weak Model**: The weak model enables knowledge transfer under privacy constraints. The weak model acts as an intermediary, allowing Hospital A to encode knowledge from its sensitive data into a form that can be shared without violating privacy laws. Only the outputs of the weak model are shared, not the raw patient data, as described in our previous response.

---

> ### Comment · Reviewer_CUwC · 2024-11-25
>
> So basically you are assuming that there exist such situations
> 1. there is a hospital that can only afford to train a small model on its own data,
> 2. and the hospital is willing to share the model's outputs to train a stronger model by someone else. Note that this strong model is only trained by the outputs of a single weak model.
>
> I believe this scenario may exist somewhere in the world but it can be quite quite rare.

---

> ### Author Response · Authors · 2024-11-27
> **Response to Reviewer CUwC**
>
> We believe this scenario is not rare, but rather the norm in the healthcare industry.
>
> Recent empirical evidence demonstrates that both conditions the reviewer identified - 1) healthcare institutions with limited AI training capabilities and 2) their willingness to engage in model-based knowledge transfer - are not rare edge cases but represent mainstream scenarios in healthcare AI adoption. According to multiple studies [1-5] resource constraints and regulatory requirements make external AI collaboration the dominant approach, not the exception.
>
> Below we provide comprehensive evidence that demonstrates how our trustworthy weak-to-strong generalization approach directly addresses these widespread, systemic challenges in healthcare AI development and deployment:
>
> - **Resource Constraints are Systemic, Not Rare**: According to the American Hospital Association, a significant portion of healthcare institutions operate under tight budgets, limiting their ability to invest in advanced computational resources [1]. This constraint is further validated by a recent MIT publication that stated hospitals face high computational and hardware costs, making local design and development of large-scale AI models unlikely [2]. The same study asserted that "healthcare organization leaders must move from facilitating internal to external solution design and development," supporting our approach of leveraging external computational resources through trustworthy weak-to-strong generalization.
>
> - **External Collaboration Under Regulatory Constraints is the Industry Standard**: Far from being rare, external collaboration in healthcare AI is becoming the dominant paradigm, shaped by both practical necessities and regulatory requirements. A 2023 BMC Digital Health study revealed that 41% of healthcare organizations primarily rely on external vendors for AI model development, with relatively few building their own solutions [3]. This same study identified “regulatory concerns” and “data security" as principal obstacles to AI adoption, highlighting that hospitals seek to develop trustworthy models without sharing their raw data. The importance of collaborative approaches is further reinforced by research in Globalization and Health, which emphasizes that "the private sector has an important role to play in initiating and scaling digital health initiatives" particularly when "resources and expertise are overstretched in the public sector" [4]. Our trustworthy weak-to-strong approach directly addresses these dual challenges by enabling healthcare organizations to leverage external institutions’ computation and technical expertise while maintaining strict control over sensitive data by only sharing weak outputs. This aligns well with the established industry pattern of seeking external AI capabilities while adhering to stringent data protection requirements.
>
> - **Global Healthcare Reality**: The relevance of our approach becomes even more apparent when considering the global healthcare landscape. According to a study in Nature Communications, resource constraints are particularly pronounced in low and middle-income countries, where hospitals lack resources to train large machine learning models effectively [5]. These institutions face multiple challenges, including "inadequate digital infrastructure" and "a shortage of skilled AI professionals." Our trustworthy weak-to-strong approach offers a practical solution by enabling these institutions to collaborate with institutions with computational resources while maintaining data sovereignty.
>
>
> These studies demonstrate that our scenario addresses fundamental challenges in healthcare AI adoption, including resource constraints, privacy regulations, and the need for trustworthy AI. The combination of lack of computational infrastructure and willingness to collaborate makes our research on trustworthy weak-to-strong generalization both relevant and timely for the healthcare industry.
>
>
> ---
>
> **References**
>
>
> [1] American Hospital Association. (2022). The Cost of Caring. https://www.aha.org/costsofcaring
>
>
> [2] Armstrong, B., Kellogg, K., Levi, R., Shah, J., & Wiesenfeld, B. (2024). Implementing generative AI in U.S. Hospital Systems. An MIT Exploration of Generative AI. https://doi.org/10.21428/e4baedd9.1729053f
>
>
> [3] Guleria, S., Guptill, J., Kumar, I. et al. Artificial intelligence integration in healthcare: perspectives and trends in a survey of U.S. health system leaders. BMC Digit Health 2, 80 (2024). https://doi.org/10.1186/s44247-024-00135-3
>
>
> [4] Labrique, A.B., Wadhwani, C., Williams, K.A. et al. Best practices in scaling digital health in low and middle income countries. Global Health 14, 103 (2018). https://doi.org/10.1186/s12992-018-0424-z
>
>
> [5] Yang, J., Dung, N.T., Thach, P.N. et al. Generalizability assessment of AI models across hospitals in a low-middle and high income country. Nature Communications 15, 8270 (2024). https://doi.org/10.1038/s41467-024-52618-6

---

> > ### Author Response · Authors · 2024-12-02
> > **Response to Reviewer CUwC**
> >
> > Dear Reviewer - we thank you again for your comments. We gave them a lot of thought and would love to hear if you feel we have satisfactorily clarified things. If so, we'd appreciate you revisiting your score or engaging in some additional discussion. Thank you!

---

> > > ### Comment · Reviewer_CUwC · 2024-12-02
> > >
> > > I think the paper would make much more sense if it considered distilling from multiple weak models to train a strong model, and the strong model would be empirically much better than each of the weak models. If this is not the case, I do not know why each hospital is willing to share its model's predictions.
> > >
> > > However, the authors disagreed with me and continued their argument on why distilling from a single weak model makes sense.
> > >
> > > I hope the paper can be properly revised in the future.

---

> > > > ### Author Response · Authors · 2024-12-03
> > > > **Response to Reviewer CUwC**
> > > >
> > > > We sincerely appreciate the reviewer’s continued engagement in our work. We would like to further address the concerns regarding the focus of our study on transferring trustworthiness from a single weak model to a strong model.
> > > >
> > > > **Importance of Single Weak Model Results**
> > > >
> > > > Our paper presents the first comprehensive evaluation of Trustworthy Weak-to-Strong Generalization (TWTS) from a single weak model to a strong model. Before we can effectively generalize to scenarios involving multiple weak models, it is crucial to establish that transferring trustworthiness is feasible and effective in the single weak model case. We shouldn't undertake the second step before confirming the viability of the first. Jumping directly to multiple weak models without understanding the single-model case could lead to overlooked challenges and less robust solutions. Demonstrating success in the one-weak-model scenario lays the essential groundwork for future explorations involving multiple weak models.
> > > >
> > > > **Multiple Weak Models Not Necessary for Impactful Collaboration**
> > > >
> > > > The applicability of our approach does not hinge on the participation of multiple hospitals. Single hospital and AI lab collaborations are both prevalent and impactful. As we stated previously, hospitals often face limitations in computational resources and AI expertise. Training large-scale, trustworthy AI models requires significant computational power and specialized knowledge that a hospital may lack. By sharing the model's predictions, the hospital can collaborate with an external AI lab to develop a stronger model. The external partner can utilize their resources to enhance performance and trustworthiness, which the hospital can then deploy for improved patient care.
> > > >
> > > > Our work directly addresses challenges in such collaborations, where the hospital may have constraints in sharing raw data due to privacy regulations and may lack the resources to train large-scale models internally. This is “why each hospital is willing to share its model's predictions” without the presence of multiple other hospitals in the collaboration.
> > > >
> > > > **Overview of Rebuttal Discussion**
> > > >
> > > > Initially, the reviewer questioned our motivation and requested a solid use case, which we provided along with five other practical scenarios where our approach is applicable. We observed that the reviewer’s focus then shifted to suggesting that our study should involve distilling from multiple weak models to a strong model. The scenario with multiple models was first introduced by us in our second batch of rebuttal responses as an additional use case to our main motivation (single weak model case). While we agree that extending our work to multiple weak models is an exciting and valuable direction for future research, our current study aims to first establish the fundamental viability of trustworthiness transfer in the single weak model case.
> > > >
> > > > **Significance of Our Contribution**
> > > >
> > > > Our paper provides the first evaluation of Weak-to-Strong Trustworthiness, offering comprehensive insights into whether trustworthiness properties, such as fairness, out-of-distribution robustness, adversarial robustness, and privacy, can be effectively transferred from a weak model to a strong model. This contribution is significant for several reasons:
> > > >
> > > > - **Practical Impact:** Many organizations, especially in high-stakes domains like healthcare, often collaborate with external partners due to resource limitations and privacy concerns. Our work addresses these real-world scenarios, providing a pathway for institutions to develop trustworthy AI models without compromising sensitive data.
> > > >
> > > > - **Comprehensive Evaluation:** We conducted extensive experiments and analyses, demonstrating the effectiveness of our approach across multiple dimensions of trustworthiness. This thorough evaluation adds robustness to our findings and offers valuable insights to the research community.
> > > >
> > > > - **Foundation for Future Work:** By thoroughly evaluating the single weak model case and establishing the feasibility of trustworthiness transfer, we set the stage for future research to build upon our findings and explore more complex scenarios involving multiple weak models.
> > > >
> > > > In conclusion, our focus on the single weak model scenario is both intentional and significant. It addresses practical, real-world situations where collaboration between a single institution and an external partner is beneficial. We acknowledge that distilling knowledge from multiple weak models to train a strong model is a promising avenue. However, exploring this extension requires first confirming that trustworthiness can be successfully transferred in the simpler case. Our current study provides this essential validation.
> > > >
> > > > We are grateful for your feedback, which has helped us strengthen our presentation and clarify the importance of our work. We hope this response addresses your concerns and demonstrates the relevance and significance of our study.

---

### Official Review · Reviewer_qE1Z · 2024-11-04

**Soundness:** 3
**Presentation:** 3
**Contribution:** 2
**Rating:** 6
**Confidence:** 3

**Summary:**

This work investigates this critical studies the transfer of trustworthiness properties from weak to strong models. This is an empirical work showing how some trustworthiness properties, such as fairness, adversarial, and OOD robustness, show significant improvement in transfer when both models were regularized towards trustworthiness, others like privacy do not exhibit signs of weak-to-strong trustworthiness.

**Strengths:**

Overall, the problem is important and well-motivated, the paper is well-written, and the experimental section had an interesting conclusion, matching the claims in abstract in intro.

**Weaknesses:**

This paper is an empirical evaluation extending a previously known result (weak-to-strong generalization) to a new set of capabilities, ie trustworthiness. Since there is no new methodology, I was hoping to see a stronger experimental section, containing a wider variety of models and datasets, ie >1 per task. I don't think the paper has any deep technical issues. In my opinion it's incomplete due to a few things, like only evaluating one definition of fairness/privacy/ood,. and using one dataset per setting. I understand using more models would be hard, due to accessibility, but if possible this would also improve the empirical evaluation.

**Questions:**

- Can you expand on the experimental setting? E.g. hyperparam search. This should be contained at least in the appendix.
- How do you connect the rate of improvement of trustworthiness capabilities with the previous results in weak-to-strong generalization?

---

> ### Author Response · Authors · 2024-11-21
> **Response to Reviewer qE1Z (Part 1)**
>
> We thank the reviewer for their clarifying questions and thoughful comments and believe that they helped improving the scope of our results. Below we address individual points raised by the reviewer
>
> > Since there is no new methodology
>
> While our paper does not propose a new algorithm, we argue that empirical insights into the capabilities and limitations of large language models (LLMs) can be as impactful as methodological innovations, especially given the rapid advancements in LLM capabilities.
>
> In particular, our work represents the *first comprehensive exploration of Trustworthy Weak-to-Strong Generalization (TWTS)*, examining how trustworthiness properties such as fairness, robustness, and privacy can be effectively transferred from weak to strong models. By introducing this novel evaluation paradigm and applying it across diverse trustworthiness criteria, we provide the community with foundational insights into a critical, underexplored problem space. This work is intended to establish a robust baseline and raise awareness of TWTS as a valuable research direction, paving the way for follow-up studies.
>
> Importantly, our two suggested methods - Weak TFT and Weak+WTS TFT - go beyond standard weak-to-strong transfer by incorporating trustworthiness regularization during different stages of training. These methods establish two strong baselines for trustworthy generalization that future research will need to surpass. This is particularly significant because our work not only evaluates existing techniques but actively provides new benchmarks to drive progress in this area. These baselines help anchor the trustworthiness generalization problem and will serve as key reference points for the community going forward.
> Furthermore, the importance of empirical work has been underscored by several prominent papers that had limited methodological innovation but were celebrated for their insightful empirical evaluations. Examples include:
>
> - Weak-to-Strong Generalization: Eliciting Strong Capabilities With Weak Supervision, ICML 2024; https://arxiv.org/abs/2312.09390
> - DecodingTrust: A Comprehensive Assessment of Trustworthiness in GPT Models, NeurIPS 2023; https://arxiv.org/abs/2306.11698
> - Chain-of-Thought Prompting Elicits Reasoning in Large Language Models, NeurIPS 2022; https://arxiv.org/abs/2201.11903
> - Rethinking the Role of Demonstrations: What Makes In-Context Learning Work?, EMNLP 2022; https://arxiv.org/abs/2202.12837
>
> These works have shown that impactful research need not always hinge on technical novelty; instead, carefully designed empirical studies addressing important gaps in understanding can be equally valuable to the community. Similarly, our work contributes by not only introducing TWTS as a research direction but also providing rigorous, well-defined baselines that set the stage for future advancements in trustworthy generalization.
>
> We hope this response clarifies the significance of our contributions and the rationale for focusing on this empirical investigation.
>
>
> > Can you expand on the experimental setting? E.g. hyperparam search. This should be contained at least in the appendix.
>
>
> Appendix A2 describes in much detail using the example of adversarial robustness how the hyperparameters were chosen. In the following, we briefly summarize our approach:
>
> Rather than relying on arbitrary or preset values, we employed a systematic empirical approach to selecting hyperparameters, focusing on finding optimal trade-offs between model performance and trustworthiness. Our methodology involved creating trade-off curves that visualize the relationship between different performance metrics (see Figure A1).
> For the adversarial robustness parameter $\lambda$, we independently fine-tuned both weak and strong models, evaluating their performance across different parameter values. We selected $\lambda^*$ as the best balance between clean and adversarial accuracy for both models.
>
> For the auxiliary loss function, we varied the $\alpha$ parameter and identified some $\alpha^*$ as the value achieving the highest accuracy across performance and trustworthiness property (see Figure A7). Similarly, we empirically determined the warm-up period for $\alpha$ and the number of fine-tuning epochs using the same trade-off curve approach.

---

> ### Author Response · Authors · 2024-11-21
> **Response to Reviewer qE1Z (Part 2)**
>
> > Since there is no new methodology, I was hoping to see a stronger experimental section, containing a wider variety of models and datasets, ie >1 per task. I don't think the paper has any deep technical issues. In my opinion it's incomplete due to a few things, like only evaluating one definition of fairness/privacy/ood,. and using one dataset per setting.
>
> **Additional datasets**. In our evaluation of weak-to-strong trends for adversarial robustness, we report accuracy values aggregated over six NLP tasks and five adversarial attacks (we have made this more clear in Appendix C):
>
> - *NLP Tasks:*
>   - SST-2 (Stanford Sentiment Treebank): A sentiment analysis task requiring classifying sentences as having positive or negative sentiment.
>   - QQP (Quora Question Pairs): A task to determine whether two questions have the same meaning.
>   - MNLI (Multi-Genre Natural Language Inference): A sentence understanding task requiring inference of entailment, contradiction, or neutrality between sentence pairs.
>    - MNLI-mm (MNLI mismatched): A variation of MNLI where validation and test data come from out-of-domain sources, increasing the challenge of generalization.
>    - QNLI (Question-answering NLI): A question-answering task framed as an entailment problem between a question and an answer candidate.
>    - RTE (Recognizing Textual Entailment): A binary entailment task with a relatively small dataset, testing generalization in low-data regimes.
>
>
> - *Adversarial Attacks*:
>   - TextBugger: Typo-based perturbations that minimally alter characters while preserving readability.
>   - TextFooler: Embedding-similarity-based perturbations that substitute words with contextually plausible alternatives.
>   - BERT-ATTACK: Context-aware perturbations leveraging BERT's language modeling capabilities to generate adversarial samples.
>   - SememePSO: Knowledge-guided perturbations that rely on semantic representations and combinatorial optimization.
>   - SemAttack: Semantic-optimization-based perturbations crafted by manipulating different semantic spaces.
>
> Aggregating the adversarial accuracy and task performance in this manner enables us to evaluate the weak-to-strong trends in a comprehensive and robust manner. The results show that our findings are consistent across a wide range of NLP tasks and adversarial attacks, indicating that they are not influenced by the specific characteristics of any single setting.
>
> **Additional metrics**. We have run additional experiments in response to the reviewer's suggestions. To address your concern, we have conducted additional experiments, this time optimizing for equalized odds in the fairness setting as suggested by [3] and measuring the membership inference attack success in the privacy setting. Notably, the results continue to exhibit trends consistent with those observed under the demographic parity objective, further supporting the robustness of our findings across different fairness metrics. These additional results demonstrate that our conclusions are not specific to a single fairness criterion and hold under more nuanced objectives like equalized odds. We incorporated these results into the revised manuscript (see Appendix B, Figures A10-A11) to provide a more comprehensive evaluation.
>
> **Additional models**. Motivated by the reviewer’s suggestions, we have run additional experiments using the Pythia 6.9B model. These results are summarized in Figure A9 of Appendix B. There, we show that our main conclusions using the larger 6.9B model do not change.
>
> We think that our efforts have significantly strengthened the evaluation in our work and hope that the reviewer agrees with us.
>
> ----
>
> We thank the reviewer again for their thoughtful comments and feedback. We hope we addressed all your questions/concerns/comments adequately. In light of our clarifications, please consider increasing your score.
>
> ----

---

> > ### Comment · Reviewer_qE1Z · 2024-11-25
> > **Response**
> >
> > The authors have addressed most of my concerns, so I'm raising the score to 6.

---

> > > ### Author Response · Authors · 2024-11-27
> > > **Response to Reviewer qE1Z**
> > >
> > > Thank you for your response. We very much appreciate that.

---

### Official Review · Reviewer_rAJ7 · 2024-11-04

**Soundness:** 3
**Presentation:** 3
**Contribution:** 3
**Rating:** 5
**Confidence:** 2

**Summary:**

In this paper, the authors explore the challenge of whether the fairness, robustness, and privacy can be transferred from weak to strong models through weak-to-strong generation. This is an important research topic, because the ability to transfer trustworthiness properties is crucial for preventing harmful outcomes, complying with regulations, and maintaining public trust in AI technologies. In this paper, the authors propose two novel approaches to enhance the transfer of trustworthiness properties between weak and strong models.

**Strengths:**

1. The authors propose two novel methods (Weak TFT and Weak+WTS TFT) to enhance trustworthiness transfer.

2. The paper is well organized and easy to follow.

3. The authors provide a comprehensive study about the related works.

4. The authors propose three-phase experimental design provides a thorough analysis of trustworthiness transfer.

**Weaknesses:**

1. The size of the strong models included in the paper is small. How about the results on the models around 7B?

2. Lack of generazation towards fairness.
	In this paper, the authors adopt the Demographic Parity as the representative definition. In fairness research domain, there are a series of fairness definition, such as Equal False Positive/Negative Rates, Calibration/Predictive Parity, etc. How the proposed method work with different fairness definition.


3. Lack the in-depth discussion about why privacy do not exhibit signs of weak-to-strong trustworthiness.


Minor:

1. Formatting issue: The subfigures in Figure 3 have varying sizes.

**Questions:**

See weakness 1, 2, 3

---

> ### Author Response · Authors · 2024-11-21
> **Response to Reviewer rAJ7**
>
> We are very grateful for the reviewer's positive and thoughful comments. Below we address individual points raised by the reviewer.
>
>
>
> > The size of the strong models included in the paper is small. How about the results on the models around 7B?
>
> Motivated by the reviewer’s suggestions, we have run additional experiments using the Pythia 6.9B model. These results are summarized in Figure A9 of Appendix B. There, we show that our main conclusions using the larger 6.9B model do not change.
>
> > In this paper, the authors adopt the Demographic Parity as the representative definition. In fairness research domain, there are a series of fairness definition, such as Equal False Positive/Negative Rates, Calibration/Predictive Parity, etc. How the proposed method work with different fairness definition.
>
> In response to the reviewer’s concern, we conducted additional experiments optimizing for equalized odds as suggested by [2]. These experiments revealed trends consistent with those observed under the demographic parity objective, indicating that our proposed method generalizes well to fairness definitions with different nuances. The results further reinforce the robustness of our findings across multiple fairness criteria.
> We have included these additional results in the revised manuscript (see Appendix B, Figure A10) to provide a more comprehensive evaluation. We hope this addresses the reviewer’s concern and highlights the flexibility and broader applicability of our method.
>
> > Lack the in-depth discussion about why privacy do not exhibit signs of weak-to-strong trustworthiness.
>
> We respectfully disagree with the reviewer’s assertion that our discussion of the privacy results lacks depth. A detailed discussion is already included in Section 4.2, specifically in the paragraph under "No TFT." However, we agree that this important point deserves greater prominence and clarity. To address this, we have added a dedicated paragraph at the end of Section 4.2 to further emphasize our insights.
>
> For the reviewer’s convenience, we reproduce the key aspects of this discussion below:
>
> *Privacy presents a unique situation. Notably, the strong ceiling does not achieve better privacy than the weak model. One key reason is that we measure the privacy property with respect to the underlying training dataset. Larger models, all else being equal, tend to memorize more information, which increases the risk of private information leakage [3]. Consequently, privacy, as measured by the extraction rate, tends to degrade when transferring knowledge from a smaller to a larger model. This occurs primarily because privacy violations in WTS-Naive are evaluated on the larger model, which is inherently more capable of memorizing its training data than the smaller model.*
>
> We believe this expanded and more prominently placed discussion will address the reviewer’s concern and provide a deeper understanding of why privacy does not exhibit signs of weak-to-strong trustworthiness in our experiment.
>
> -------
>
> We thank the reviewer again for their thoughtful comments and feedback. We hope we addressed all your questions/concerns/comments adequately. In light of our clarifications and since the reviewer rated 'soundness', 'presentation' and 'contribution' as 'good', please consider increasing your score.
>
> --------
> **References**
>
> [1] Weak-to-Strong Generalization: Eliciting Strong Capabilities With Weak Supervision, https://openai.com/index/weak-to-strong-generalization/, ICML 2024
>
> [2] Achieving Equalized Odds by Resampling Sensitive Attributes, https://proceedings.neurips.cc/paper/2020/hash/03593ce517feac573fdaafa6dcedef61-Abstract.html, NeurIPS 2020
>
> [3] Gaussian membership inference privacy. https://arxiv.org/abs/2306.07273, NeurIPS, 2024,

---

> > ### Author Response · Authors · 2024-12-02
> > **Response to Reviewer rAJ7**
> >
> > Dear Reviewer, we thank you again for your comments and detailed feedback. We gave them a lot of thought and would love to hear what you think about our rebuttal and if you have any follow-ups. If you feel we have satisfactorily clarified things we'd appreciate you revisiting your score or engaging in some discussion about the paper. Thank you

---

### Official Review · Reviewer_pMn1 · 2024-11-08

**Soundness:** 3
**Presentation:** 3
**Contribution:** 2
**Rating:** 6
**Confidence:** 3

**Summary:**

In this paper the authors studied the abiilty for a weak model to teach a more powerful model to be fair, robust (both OOD and adversarial), and privacy sensitive.  The paper explores two appraoches: (1) traditional weak-to-strong generalization after training the weak model to have good trustworthiness and (2) additional trustworthiness regularization of the strong model. They find that traditional weak-to-strong generalization does not hold but that their second method works better.

**Strengths:**

* While weak-to-strong generalization has been studied as a general property, it has not been sufficiently examined for safety and trustworthiness properties.

* The paper is fairly clear and thorough in its experiments, making it relatively easy to interpret.

**Weaknesses:**

* Some of the tasks chosen I don't believe are a good fit for the study.  As an example, classification over discrete features (as in Adult) is an odd fit for testing LLMs (in contrast to something like BBQ) and demographic parity as an objective is particularly easy for weak models to learn such that teaching the concept to stronger models doesn't require much. I'd expect the weak-to-strong hypothesis is more interesting in tasks where it is hard for weak models and larger models we think can be more capable but giving them good supervision is hard.  I'd expect this is true in more nuanced definitions of fairness, stereotyping, and safety as an example.  (Similar statements could be said for newer ways of evaluating privacy and robustness to jailbreaks)

* There is some lack of clarity in the method description here so I could be wrong but it seems like the Weak+WTS TFT method is largely succeeding by adding the trustworthiness regularization directly and maybe not needing the weak model at all?  My current reading as a result is that the paper is largely presenting instead a negative result for WTS genrealiztaion on trustworthiness.

* There are some missing details that I think significantly shape the results, eg what is the actual loss for Weak+WTS TFT, what data is used for every method, and what is the vanilla form of using each type of data.

**Questions:**

* Am i understanding correctly that in Weak+WTS TFT the strong model gets trustworthiness regularization + the normal weak-to-strong loss?  does the trustworthiness regularizer make use of the weak model?
* How is the Enron data used?
* Is all fine-tuning only on the trustworhtiness datasets? What is the vanilla form of each task that doesn't apply the trustworthiness loss?
* Is it a reasonable assumption that we have these curated datasets? It would seem that even having some of the datasets makes the WTS problem secondary (eg once I know demographic attributes and have decided on demogrpahic parity, or differential privacy)

nits:
* "modelss" (extra s)

---

### Author Response · Authors · 2024-11-21
**Summary of Revisions**

We thank all the reviewers for their valuable feedback, which has significantly contributed to the refinement of our work. Inspired by their comments and suggestions, we are providing the following changes to our work:

- **Clarified Contributions**: Considering reviewers ``qE1Z``’s and ``CUwC``’s feedback, we have updated the abstract and introduction to state our contributions more clearly.

- **Added detail for all three training objectives**: Addressing Reviewer ``pMn1``’s feedback, we have updated Sections 3.1 and 3.2 and added a Section to Appendix A.1 to give more details on the training objectives.

- **Added overview of main conclusions from experiments in Table 1**:  In response to all reviewers, we have added Table 1 to the manuscript which provides a concise overview of the weak-to-strong trustworthiness trends observed in our experiments.

- **Added additional Experiments**: Inspired by the constructive suggestions from Reviewers ``pMn1``, ``rAJ7`` and ``qE1Z``, we have conducted additional experiments to showcase the effect of:

   - **Larger Model Size (6.9B) on Weak-to-strong Trustworthiness**: In response to Reviewer ``rAJ7``, we ran additional experiments for fairness, adversarial robustness, and OOD robustness using the larger 6.9B Pythia LLM. The conclusions are qualitatively not different from the conclusions from the 410M and 1B large models.

   - **Additional Data sets**: Responding to Reviewer ``qE1Z``, we would like to clarify that the Adv. Robustness results are averaged across 6 datasets. Overall, the trends on each dataset are not different from the global trends we observe Adv. Robustness.

   - **Additional Evaluation Measures**: In response to Reviewer ``pMn1``’s and ``rAJ7``’s suggestions, we have added additional experimental results for fairness evaluation (Equalized Odds) and privacy (Membership Inference Attacks).  Here, the trends on each evaluation measure is consistent with the trend we observed before.

----
We would additionally like to highlight that our work makes several *impactful contributions*:
- **Novel Trustworthy Learning Paradigm**: This is the first work to investigate if trustworthiness properties can generalize from a weak to a strong model using weak-to-strong supervision, a process we term weak-to-strong trustworthiness generalization.

- **Foundational Training Strategies for Weak-to-Strong Trustworthiness Generalization**: We introduce two strong baseline training strategies,
``Weak TFT`` and ``Weak+WTS TFT``, designed to facilitate weak-to-strong trustworthiness generalization.

- **Weak-to-Strong Trustworthiness Generalization is Feasible**: Our experiments show that some trustworthiness properties can indeed be generalized and even enhanced from weak to strong models.

- **Comprehensive Evaluation**: We perform extensive experiments across:
   - *Models*: Ranging from weak (14M, 70M, 160M) to strong (410M, 1B, 7B) sizes.
   - *Tasks*: Assessing fairness, privacy, adversarial robustness, and out-of-distribution (OOD) robustness.
   - *Datasets & Metrics*: Spanning 9 datasets, the two most popular privacy attack metrics (Membership Inference, Extraction), two most popular fairness criteria (Equalized Odds, Demographic Parity), and standard robustness metrics (Adv & OOD robustness)
   - *Thorough Hyperparameter Optimization*: We conduct exhaustive and computationally expensive hyperparameter searches for all relevant parameters in the objective functions to ensure robust and fair comparisons.

---

### Meta-Review · Area_Chair_r7hD · 2024-12-17

**Metareview:**

The paper explores the transfer of trustworthiness properties—fairness, robustness (OOD and adversarial), and privacy—from weak models to stronger ones via weak-to-strong generalization. While the study is well-organized, addressing an important topic with comprehensive experiments, its contributions are limited. The experimental design focuses on a single fairness definition (Demographic Parity) and a narrow range of datasets and model scales, limiting generalizability. Additionally, the results suggest that trustworthiness regularization alone, rather than weak-to-strong generalization, drives the improvements, undermining the paper's central premise. The lack of discussion on the failure of privacy transfer, combined with missing details on the experimental methods (e.g., loss functions, data usage), further weakens the work. Overall, while the topic is relevant, the lack of technical novelty, limited scope, and incomplete exploration make this paper insufficient for acceptance.

**Additional Comments On Reviewer Discussion:**

There is quite an extensive discussion between Reviewer CuWC and the authors on the motivation of the work, basically, Reviewer CuWC does not think the problem setting is practical and ubiquitous. I think this raises an important limitation of this work, which is not well addressed during the rebuttal.

---

### Decision · Program_Chairs · 2025-01-22

Reject